# A *Shigella sonnei* clone with extensive drug resistance associated with waterborne outbreaks in China

Shaofu Qiu[1,12], Kangkang Liu [1,12], Chaojie Yang[1,12], Ying Xiang[1,12], Kaiyuan Min [2,12], Kunpeng Zhu[1], Hongbo Liu[1], Xinying Du[1], Mingjuan Yang[1], Ligui Wang[1], Yong Sun[3], Haijian Zhou[4], Muti Mahe[5], Jiayong Zhao[6], Shijun Li[7], Deshan Yu[8], Jane Hawkey [9], Kathryn E. Holt[9], Stephen Baker[10], Juntao Yang [2] ✉, Xuebin Xu [11] ✉ & Hongbin Song [1] ✉

Antimicrobial resistance of *Shigella sonnei* has become a global concern. Here, we report a phylogenetic group of *S. sonnei* with extensive drug resistance, including a combination of multidrug resistance, coresistance to ceftriaxone and azithromycin (cef^R azi^R), reduced susceptibility to fluoroquinolones, and even colistin resistance (col^R). This distinct clone caused six waterborne shigellosis outbreaks in China from 2015 to 2020. We collect 155 outbreak isolates and 152 sporadic isolates. The cef^R azi^R isolates, including outbreak strains, are mainly distributed in a distinct clade located in global Lineage III. The outbreak strains form a recently derived monophyletic group that may have emerged circa 2010. The cef^R azi^R and col^R phenotypes are attributed to the acquisition of different plasmids, particularly the IncB/O/K/Z plasmid coharboring the $bla_{CTX-M-14}$, *mphA*, *aac(3)-IId*, *dfrA17*, *aadA5*, and *sul1* genes and the IncI2 plasmid with an *mcr-1* gene. Genetic analyses identify 92 accessory genes and 60 single-nucleotide polymorphisms associated with the cef^R azi^R phenotype. Surveillance of this clone is required to determine its dissemination and threat to global public health.

Shigellosis, caused by species of the *Shigella* genus, remains an important public health threat globally. Historical data suggest that *Shigella* spp. cause approximately 165 million cases of shigellosis annually, resulting in more than one million deaths worldwide[1], with the majority occurring in children in low-income countries[2]. The *Shigella* genus is composed of four species: *Shigella dysenteriae*, *Shigella flexneri*, *Shigella boydii*, and *Shigella sonnei*. Historically, *S. sonnei* is a major contributor to the global burden of dysenteric diarrhea in developed countries, while it is now undergoing an unprecedented expansion across industrializing regions, particularly in Asia[3,4]. In China, a rising dominance of *S. sonnei* has been observed, with *S. sonnei* replacing *S. flexneri* as the predominant species[5]. Therefore, *S. sonnei* is

[1]The Chinese PLA Center for Disease Control and Prevention, Beijing, China. [2]State Key Laboratory of Medical Molecular Biology, Institute of Basic Medical Sciences, Chinese Academy of Medical Sciences & Peking Union Medical College, Beijing, China. [3]Anhui Provincial Center for Disease Control and Prevention, Hefei, China. [4]National Institute for Communicable Disease Control and Prevention, Chinese Center for Disease Control and Prevention and State Key Laboratory for Infectious Disease Prevention and Control, Beijing, China. [5]Center for Disease Control and Prevention of Xinjiang Uygur Autonomous Region, Urumqi, China. [6]Henan Provincial Center for Disease Control and Prevention, Zhengzhou, China. [7]Guizhou Provincial Center for Disease Control and Prevention, Guiyang, China. [8]Gansu Provincial Center for Disease Control and Prevention, Lanzhou, China. [9]Department of Infectious Diseases, Monash University, Melbourne, VIC, Australia. [10]University of Cambridge School of Clinical Medicine, Cambridge Biomedical Campus, Cambridge, United Kingdom. [11]Shanghai Municipal Center for Disease Control and Prevention, Shanghai, China. [12]These authors contributed equally: Shaofu Qiu, Kangkang Liu, Chaojie Yang, Ying Xiang, Kaiyuan Min. ✉e-mail: yangjt@pumc.edu.cn; xxb72@sina.com; hongbinsong@263.net

emerging globally as the dominant causative agent of bacterial dysentery.

Antimicrobials, especially fluoroquinolones (e.g., ciprofloxacin and norfloxacin), third-generation cephalosporins (e.g., ceftriaxone), and macrolides (e.g., azithromycin), are recommended by the World Health Organization (WHO) for the treatment of shigellosis to accelerate recovery, prevent complications, and reduce onward transmission[6]. Unfortunately, a pandemic clone of multidrug-resistant (MDR) S. sonnei has been found to be globally distributed in recent years, and this pandemic clone is located within global Lineage III, which recently emerged from Europe and underwent contemporary global dispersal and localized clonal expansion[3,7]. Accordingly, the WHO and US Centers for Disease Control and Prevention (CDC) have declared MDR Shigella a serious threat[2,8]. Moreover, resistance to fluoroquinolones has been increasingly detected among MDR S. sonnei, and studies have shown that South Asia acts as a reservoir for a single clone of ciprofloxacin-resistant S. sonnei, which has caused travel-associated infections and even large outbreaks in many countries[9,10].

In 2012, Holt et al. performed phylogenetic analysis on 132 globally distributed S. sonnei isolates, and the resulting phylogeny was divided into four distinct lineages (Lineage I to Lineage IV)[3]. Several studies have investigated the genomic epidemiology of S. sonnei at the global and regional levels, including in Asia, Australia, the Middle East, South America, and the UK[3,7,9,11–13], and have identified additional phylogenetic groups of Lineage III with great public health significance. These studies on S. sonnei strains from the UK and France[12], Latin America and the Caribbean[11], Vietnam[7], Bhutan, Australia, and Ireland[9] showed that most of these newly disseminated S. sonnei groups belonged to a single, globally distributed, multidrug-resistant clade of S. sonnei Lineage III, which was referred to as global Lineage III. Furthermore, these groups are associated with features such as ciprofloxacin resistance, azithromycin resistance, and transmission among men who have sex with men (MSM)[9,12,13]. These results reveal that Lineage III has become a popular branch of MDR S. sonnei, posing a great threat to public health in these countries and even around the world. We reported a waterborne shigellosis outbreak caused by MDR S. sonnei with resistance to ceftriaxone and azithromycin (cef$^R$azi$^R$) that occurred in 2015[14]. Since then, we have identified the prevalence of MDR S. sonnei with cef$^R$azi$^R$ and reduced susceptibility to fluoroquinolones, which has caused five additional waterborne shigellosis outbreaks in China. Surprisingly, some of the outbreak isolates had even acquired a plasmid-mediated mcr-1 gene, conferring resistance to colistin (col$^R$). The emergence and prevalence of MDR S. sonnei isolates with concurrent resistance to ciprofloxacin, azithromycin, ceftriaxone, and/or colistin will inevitably greatly narrow the choice of effective antimicrobials, especially among children (considering the cartilage toxicity of fluoroquinolones, ceftriaxone and azithromycin are recommended as alternative treatments for children[15]), and thus will pose a more serious threat to global public health.

In this work, we perform whole-genome sequencing (WGS) of the outbreak and sporadic S. sonnei isolates with this distinct resistance profile to determine the origin, spread, and local establishment of these extensively drug-resistant (XDR) S. sonnei isolates. We find that most of these isolates are located in Lineage III and form a distinct clade. The cef$^R$azi$^R$ isolates, including the outbreak strains, are mainly distributed in this clade and the outbreak strains form a recently derived monophyletic group that may have emerged circa 2010. The cef$^R$azi$^R$ and col$^R$ phenotypes are attributed to the acquisition of different plasmids, particularly the IncB/O/K/Z plasmid coharboring the bla$_{CTX-M-14}$, mphA, aac(3)-IId, dfrA17, aadA5, and sul1 genes and the IncI2 plasmid with an mcr-1 gene. Ninety-two accessory genes and 60 single-nucleotide polymorphisms (SNPs) are associated with the cef$^R$azi$^R$ phenotype, supporting the divergent evolution of S. sonnei via the accumulation of genomic signatures. Heavy precipitation and floods may be relevant to the prevalence of the XDR clone. Continued surveillance and further genomic epidemiological studies are urgently required to determine the potential dissemination and consequent risk to global public health of this clone.

## Results

### Characteristics of six shigellosis outbreaks in China

We investigated six shigellosis outbreaks in China and summarized their characteristics. Among the outbreaks, five occurred in Guangxi Province and one in Anhui Province, which are all located in southern China (Fig. 1a). The outbreaks occurred mainly in kindergartens and primary and middle schools. From these outbreaks, 433 patients were confirmed (Supplementary Table S1). The majority (92.4%) of these patients were under 14 years old, and male patients accounted for 48.3%. They all showed symptoms of diarrhea, followed by fever (94.0%), abdominal pain (70.2%), vomiting (67.7%), rectal tenesmus (50.3%), nausea (22.2%), and dizziness (1.6%) (Supplementary Table S1). All these patients recovered, with no deaths.

A total of 378 samples were collected from these outbreaks, including 237 stool samples, 19 rectal swabs, 2 vomitus samples, 53 residual food samples, and 67 water samples; 155 S. sonnei isolates were identified, including 11, 18, 40, 26, 22, and 38 strains from the Yulin outbreak, Baise outbreak, Nanning outbreak, Guigang outbreak, Wuzhou outbreak, and Huainan outbreak, respectively. Among the 67 water samples, 6, 34, 5, 6, and 16 were collected from the Yulin outbreak, Baise outbreak, Nanning outbreak, Guigang outbreak, and Huainan outbreak, respectively; six isolates were recovered from water samples, including two (one from tap water and one from well water) from the Yulin outbreak, one (from tap water) from the Baise outbreak, two (one from tap water and one from well water) from the Nanning outbreak, and one (from tap water) from the Guigang outbreak. Three of the 16 water samples from the Huainan outbreak tested positive for Shigella nucleic acids by real-time polymerase chain reaction assays, although S. sonnei isolates were not recovered.

Investigations on water supply systems were performed for five of the six outbreaks (all except for the Wuzhou outbreak). For the locations of the Yulin outbreak, Nanning outbreak, and Guigang outbreak, water supplies were sourced from self-provided reservoirs pumped from the digging wells. For the Baise outbreak location, a centralized water supply sourced from a mountain spring water well was used, and for the Huainan outbreak location, a centralized water supply sourced from a waterworks facility was used. All the water sources were not well protected, and the water supplies serving all people were not strictly chlorinated and lacked complete sterilization records. The water-quality indices, including the aerobic bacterial count and total coliform group, exceeded the standards.

### Antimicrobial resistance of S. sonnei isolates

In addition to the 155 outbreak strains, we also recovered 152 sporadic strains from historical routine surveillance of shigellosis. The sporadic strains were isolated from patients infected with S. sonnei, mainly during 2004-2016 (Figs. 2, 1b). We carried out a microbiological investigation into both the outbreak and the sporadic strains. Antimicrobial susceptibility tests showed that the outbreak isolates, including 149 isolates from patients and six from water samples, all had the cef$^R$azi$^R$ phenotype, with additional resistance to ampicillin, ticarcillin, piperacillin, trimethoprim/sulfamethoxazole, gentamicin and tetracycline (Supplementary Table S2). In addition, 30 strains from the Nanning outbreak showed alarming resistance to colistin. Among the 152 sporadic isolates, 32.9% had the cef$^R$azi$^R$ phenotype. Overall, the outbreak strains showed higher rates of resistance to the abovementioned antimicrobials than the sporadic Chinese strains (Supplementary Table S2). Real-time polymerase chain reaction screening and Sanger sequencing showed that all of the outbreak strains harbored

the $bla_{CTX-M-14}$ and *mphA* genes, conferring cef$^R$azi$^R$ resistance, and the col$^R$ strains carried the *mcr-1* gene. S1 nuclease pulsed-field gel electrophoresis (S1–PFGE) and Southern blotting showed that the $bla_{CTX-M-14}$ and *mphA* genes were colocalized on the -100 kbp plasmid, and the *mcr-1* gene was located on the -60 kbp plasmid (Supplementary Fig. S1).

## Phylogenetic analysis of *S. sonnei* isolates

We drew a maximum-likelihood phylogenetic tree based on 462 genomes (155 outbreak isolates, 152 sporadic isolates, and 155 globally distributed isolates described previously[3,12], as detailed in Supplementary Table S3) to investigate the population structure and evolutionary position of the Chinese *S. sonnei* isolates, including the cef$^R$azi$^R$ isolates, within the global *S. sonnei* phylogeny. The phylogenetic tree (Fig. 1c) showed that the 462 genomes were divided into four distinct lineages as described by Holt KE et al.[4], and the Chinese isolates were distributed within two of the four lineages (Lineage I and Lineage III). The majority of the Chinese isolates were located in Lineage III and formed a distinct Chinese clade (main Chinese clade). Further genotyping of the main Chinese clade strains showed that they belonged to the 3.7.6 genotype, corresponding to global Lineage III (Fig. 1c). Multilocus sequence typing (MLST) analysis showed that all of the strains belonged to ST152, as described by Hawkey et al.[13] It is estimated that the main Chinese *S. sonnei* clade shares a most recent common ancestor (MRCA) that existed circa 1987 (Fig. 1c), suggesting a recently derived clone of *S. sonnei* circulating in China. The cef$^R$azi$^R$ isolates were mainly distributed in the main Chinese clade, which was traced back to 1999 by Bayesian phylogenetic analysis (Fig. 1c). The outbreak isolates formed a recently derived monophyletic group that may have emerged circa 2010 (Figs. 1c, 2).

## Microevolution of cef$^R$azi$^R$ *S. sonnei* isolates

As shown in Supplementary Table S3, our dataset contains genome sequences with different spatiotemporal and phenotypic contexts, offering an opportunity to explore the microevolution and phylogeography of cef$^R$azi$^R$ isolates, especially the outbreak strains, within China. A total of 215 cef$^R$azi$^R$ isolates were screened, including 205 Chinese cef$^R$azi$^R$ isolates and ten UK cef$^R$azi$^R$ isolates. The Chinese cef$^R$azi$^R$ isolates were recovered from nine of the 11 monitored regions of China from 2008 to 2020 (Supplementary Table S3). The Shanghai isolates were widely distributed among the cef$^R$azi$^R$ groups (Fig. 2), revealing Shanghai as a hub for the transmission of cef$^R$azi$^R$ isolates within China. A total of 159 SNPs were identified among the six outbreak isolates, and 7, 18, 20, 7, 9, and 17 SNPs were identified among the Yulin, Baise, Nanning, Guigang, Wuzhou, and Huainan outbreaks, respectively (Fig. 2). The strains in the Huainan outbreak were closer to those in the Yulin outbreak, with 46 SNPs in the 49 strains. The strains from Baise were closer to those from Guigang, with 22 SNPs in 44 strains, and the strains from Nanning were closer to those from Wuzhou, with 32 SNPs among the 62 strains. The geographical locations of the five outbreaks in Guangxi from west to east are in the order Baise, Nanning, Guigang, Yulin, and Wuzhou (Fig. 1a and Supplementary Fig. S2). These five cities are located in the Pearl River Basin, and the Pearl River flows from southwest to northeast. The location of Huainan is near the Yangtze River Basin and Huaihe Basin. Moreover, in the Yulin, Baise, Nanning, and Guigang outbreaks, the cef$^R$azi$^R$ isolates recovered from water samples and outbreak patients were similar, with only one to three different SNPs, suggesting that the contaminated water supply may have been the cause of the outbreaks. These results indicated that the cef$^R$azi$^R$ isolates have been prevalent in diverse regions of China, undergone clonal expansion, and then

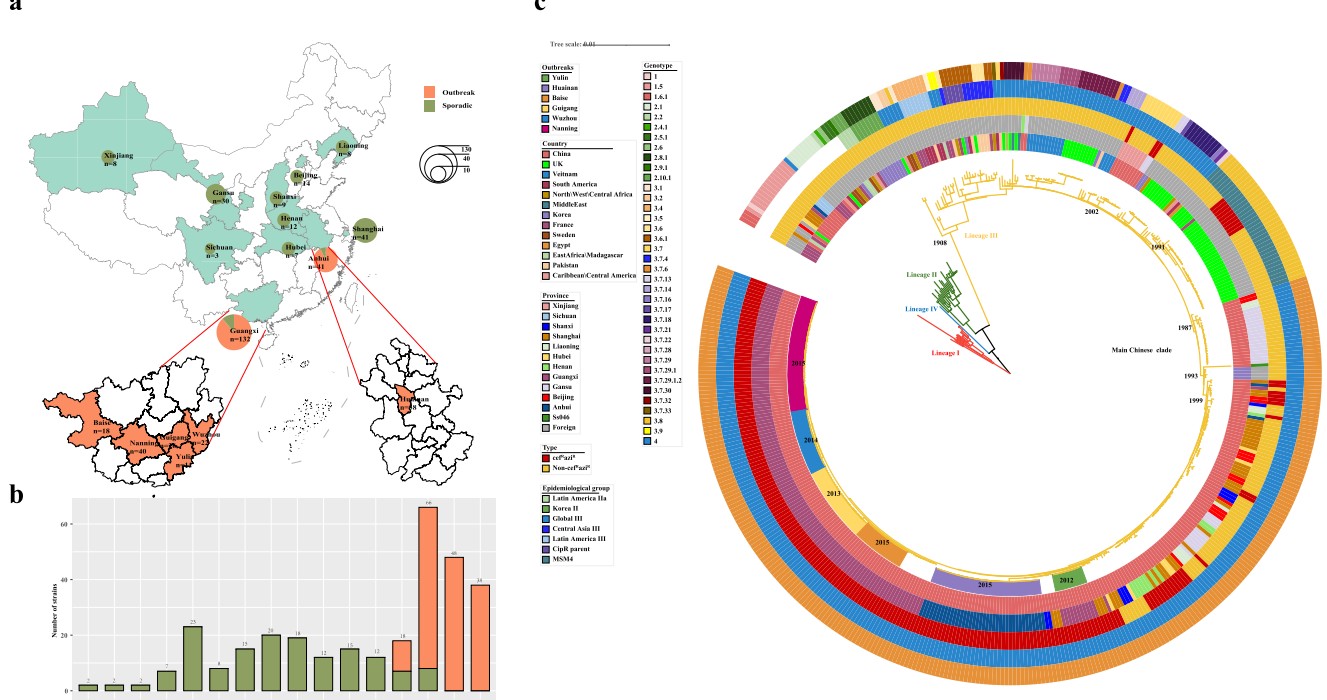

**Fig. 1 | Distribution and phylogenetic analysis of Chinese *S. sonnei* strains, including the outbreak strains. a** shows the regional distribution of Chinese *S. sonnei* isolates, including sporadic and outbreak strains. There were 307 Chinese strains, including 155 outbreak strains and 152 sporadic strains from different regions of China. **b** shows the temporal distribution of Chinese *S. sonnei* isolates, including sporadic and outbreak strains. The outbreak strains were collected in 2015, 2016, 2019, and 2020. **c** shows a phylogenetic tree of global *S. sonnei* strains, displaying the position of Chinese *S. sonnei* strains, including the cef$^R$azi$^R$ isolates as well as the six outbreak strains. The rings, from outer to inner, labeled with different colors indicate the cef$^R$azi$^R$ and non-cef$^R$azi$^R$ phenotypes, the isolation regions, geographical origin, and outbreak locations, respectively. Divergence dates are given on the major nodes in the tree. The map was created using publicly available data (https://www.webmap.cn/main.do?method=index). Source data are provided as a Source Data file. cef$^R$azi$^R$ coresistance to ceftriaxone and azithromycin.

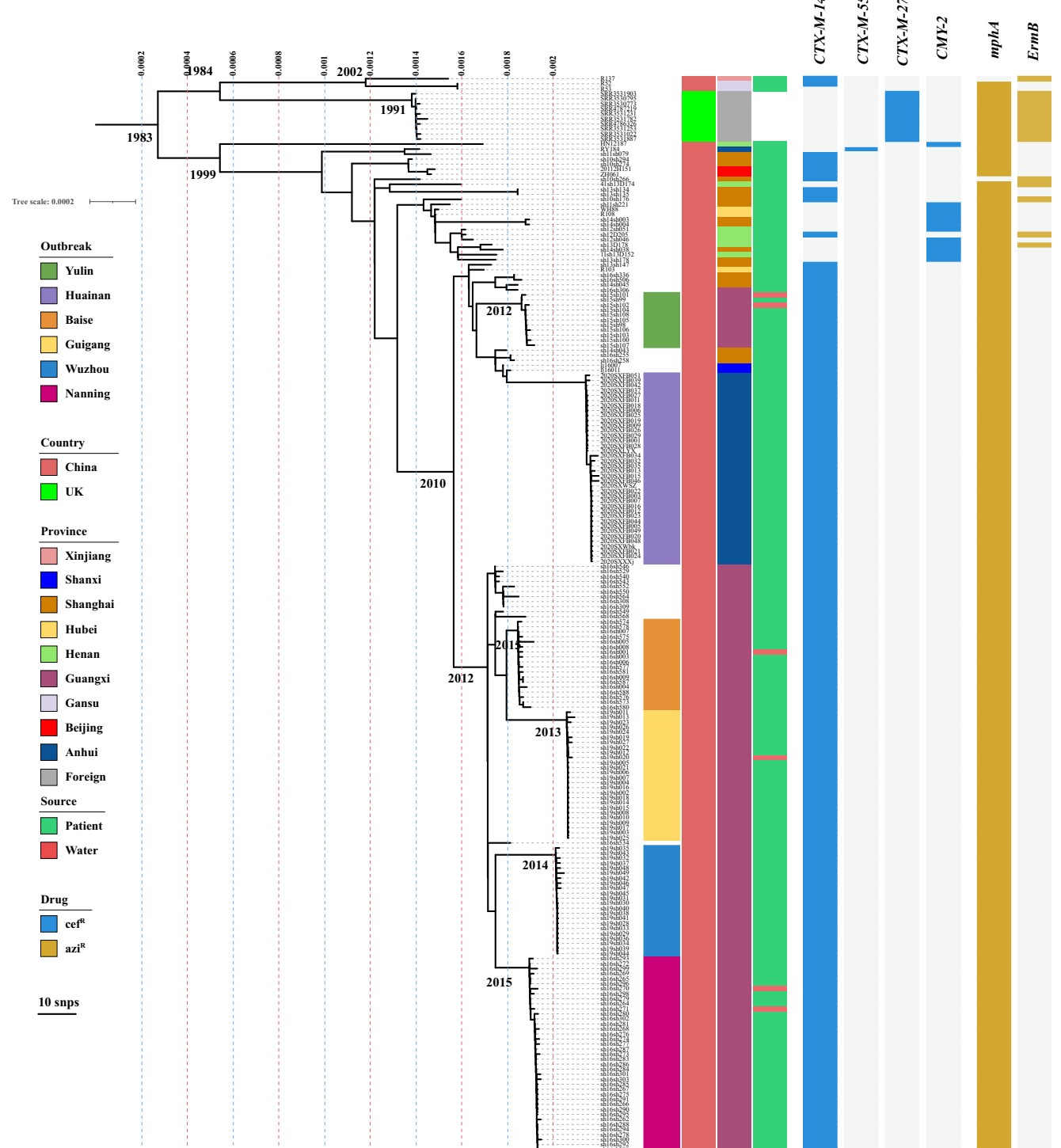

**Fig. 2 | Microevolution of the cef$^R$azi$^R$ *S. sonnei* strains including the outbreak strains.** The phylogenetic tree was constructed based on 215 cef$^R$azi$^R$ *S. sonnei* isolates, including 205 Chinese strains and 10 UK strains. The outbreak locations, isolation regions, geographical origin, and isolation sources (patient or water) are labeled in different colors. The AMR genes conferring cef$^R$azi$^R$ coresistance are shown with a heatmap. Source data are provided as a Source Data file. cef$^R$ resistance to ceftriaxone, azi$^R$ resistance to azithromycin, SNP single-nucleotide polymorphism, cef$^R$azi$^R$ coresistance to ceftriaxone and azithromycin, AMR antimicrobial resistance.

caused shigellosis outbreaks, and the outbreak clone probably originated from an MRCA.

**Antimicrobial resistance determinants among *S. sonnei* isolates**

The acquisition of antimicrobial resistance (AMR) determinants may act as a major driving force for the adaptive evolution of the MDR

*S. sonnei* lineages. Genomic investigation of acquired AMR genes showed that the four lineages contained different numbers of AMR genes, and the isolates in Lineage III contained more AMR genes than those in other lineages (Supplementary Fig. S3A and Fig. 3). The isolates in Lineage III carried an average of 9.9 AMR genes, containing mostly *aac(3)-IIa*, *aadA1*, *aadA5*, *sat-2*, *strA*, and *strB* conferring

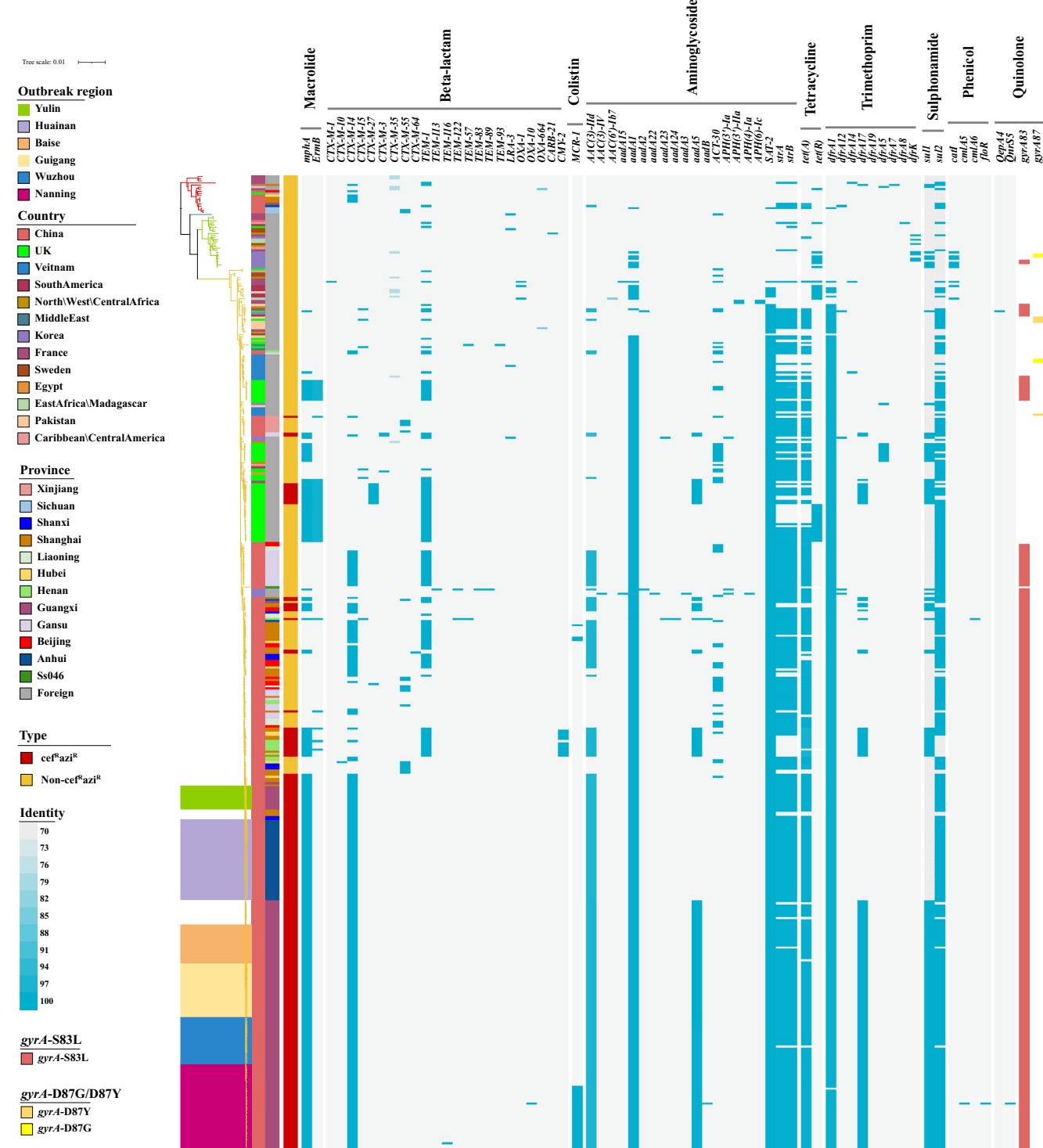

**Fig. 3 | Phylogeny of the global *S. sonnei* isolates with a heatmap showing the distribution of AMR determinants.** The region covered by color patches on the evolutionary tree represents the outbreak strains in different regions, and the two bands at the left of the heatmap indicate the information of the country and province of the strains, respectively. Blue stripes represent the identity of different types of AMR genes, showing genes with only more than 90% coverage and 70% identity. GyrA mutations, including S83L, D87Y, and D87G, are separately indicated by the colors red, gamboge, and yellow. Source data are provided as a Source Data file. cefʳaziʳ coresistance to ceftriaxone and azithromycin, AMR antimicrobial resistance.

resistance to aminoglycosides, *tet(A)* conferring resistance to tetracycline, *dfrA1* conferring resistance to trimethoprim, and *sul1* and *sul2* conferring resistance to sulfonamide[3,7]. The cefʳaziʳ isolates including the outbreak isolates had more AMR genes than the non-cefʳaziʳ isolates (Supplementary Fig. S3B, C), with an average of 11.7 AMR genes, carrying mostly the *bla*CTX-M-14, *mphA*, *aac(3)-IId*, *aadA1*, *aadA5*, *sat-2*,

*strA*, *strB*, *tet(C)*, *dfrA1*, *dfrA17*, and *sul2* genes. We found that there was an *intI1*-type integron located downstream of the *dfrA17* gene associated with the cluster of *dfrA17*, *aadA5*, *qacEdelta*, and *sul1* (Supplementary Fig. S4). In addition, there were four other gene clusters associated with integrons. The genes *tet(A)*, *strA*, *strB*, and *sul2* were mainly present in the order *sul2*, *tet(A)*, *strB*, and *strA*, with a 135 bp *intI1*

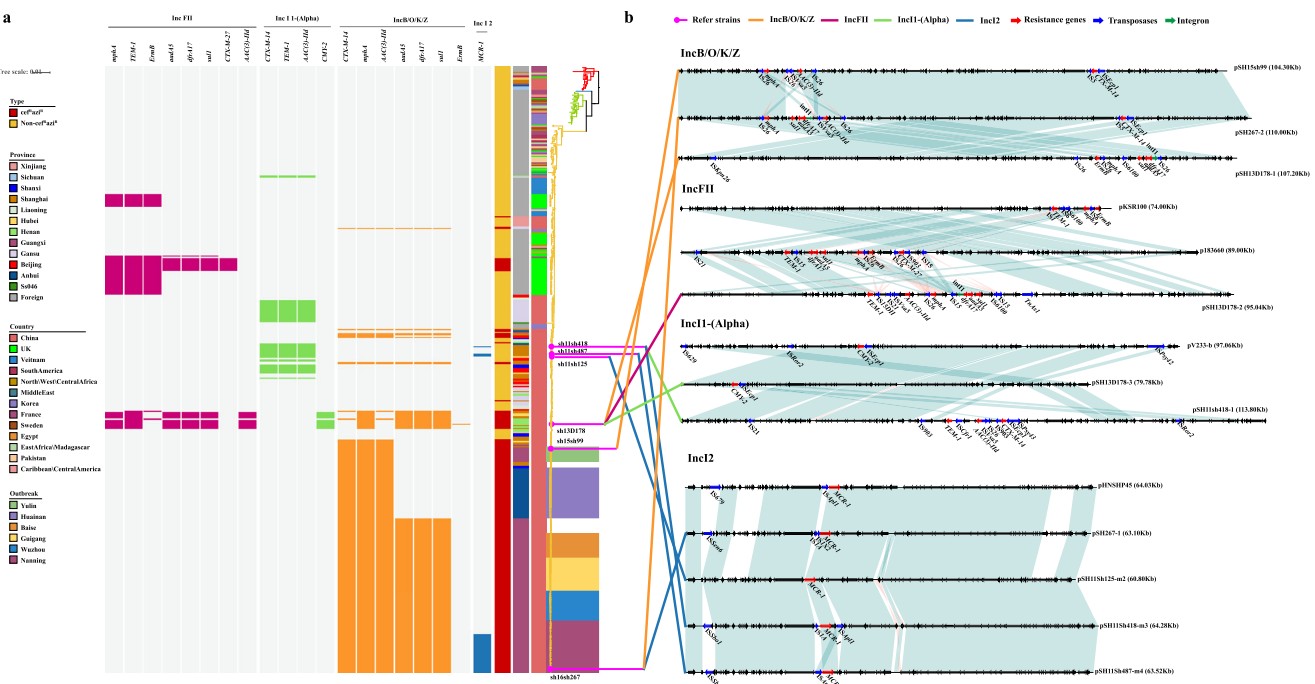

**Fig. 4 | Plasmid profiles and pairwise comparisons of plasmids associated with the cef^R azi^R and col^R phenotypes. a** Plasmids and plasmid-mediated resistance genes are represented in the heatmap by red, green, yellow, and blue bands, respectively. **b** Pairwise comparisons of the plasmids harboring the $bla_{CTX-M}$, $bla_{CMY-2}$, $bla_{TEM}$, $mphA/ermB$, and $mcr-1$ genes. Four replicons of plasmids, including pSH15sh99, pH267-2, and pSH13D178-1 of IncB/O/K/Z; pKSR100, p183660, and pSH13D178-2 of IncFII; pV233-b, pSH13D178-3, and pSH11sh418-1 of IncI1-(Alpha); and pHNSHP45 and pSH267-1 of IncI2, were selected for pairwise comparisons. Source data are provided as a Source Data file. cef^R azi^R coresistance to ceftriaxone and azithromycin, col^R resistance to colistin.

fragment between the $tet(A)$ and $strB$ genes. Another combination of the $sul1$, $qacEdelta$, and $aadA1$ genes had an $intI1$ downstream of $aadA1$. The genes $dfrA1$, $sat-2$, and $aadA1$ were arranged in the order $aadA1$, $sat-2$, and $dfrA1$ with an $intI2$ integron downstream of the $dfrA1$ gene. There were still some strains carrying the $dfrA1$ and $sat-2$ genes with $intI2$ downstream of $dfrA1$ (Supplementary Fig. S4). Among the isolates with the cef^R azi^R phenotype, the outbreak isolates contained a similar number of AMR genes as the sporadic cef^R azi^R isolates (Supplementary Fig. S3D). Compared with the isolates from the Yulin and Huainan outbreaks, the isolates from the Baise, Nanning, Guigang, and Wuzhou outbreaks acquired additional $aadA5$, $sul1$, and $dfrA17$ genes, and surprisingly, the isolates from the Nanning outbreak acquired the $mcr-1$ gene (Fig. 3). Remarkably, the isolates from the main Chinese clade, including the outbreak isolates, all carried an S83L mutation in the $gyrA$ gene that confers resistance to nalidixic acid and reduced susceptibility to fluoroquinolones (Fig. 3).

### Profiles and comparisons of plasmids associated with cef^R azi^R and col^R phenotypes

To determine the molecular mechanisms conferring cef^R azi^R and col^R resistance, four plasmids representing different replicons harboring $bla_{CTX-M}$, $bla_{CMY-2}$, $bla_{TEM}$, $mphA/ermB$, or $mcr-1$ were selected for further analysis, namely, the IncB/O/K/Z, IncFII, IncI1-(Alpha), and IncI2 plasmids (Fig. 4). With the Illumina and Nanopore sequencing platforms, 31 complete plasmids were obtained in this study, and these plasmids harbored one or more of the following resistance genes: $bla_{CTX-M}$, $bla_{CMY-2}$, $bla_{TEM}$, $mphA/ermB$, and $mcr-1$ (Supplementary Table S4). Analysis of the assembled contigs revealed that a small clone of the UK cef^R azi^R isolates contained an IncFII plasmid designated p183660 (GenBank accession no. KX008967 [https://www.ncbi.nlm. nih.gov/nuccore/KX008967.1/])[16], which carried the $bla_{CTX-M-27}$, $bla_{TEM-1}$, $mphA$, $ermB$, $dfrA17$, $aadA5$ and $sul1$ genes (Fig. 4a, b). Unlike the UK cef^R azi^R isolates, most of the Chinese cef^R azi^R isolates, including all of the outbreak isolates, carried a plasmid with close homology to

pSH15sh99 (GenBank accession no. KY471628 [https://www.ncbi.nlm. nih.gov/nuccore/KY471628.1/]) (Fig. 4a, b), a 104,285 bp IncB/O/K/Z plasmid from the cef^R azi^R isolates from the Yunlin outbreak[14]. Similar to pSH15sh99, these IncB/O/K/Z plasmids carried an IS26 transposase upstream of $mphA$ and ISEcp1 upstream of $bla_{CTX-M-14}$. However, compared with pSH15sh99, the plasmids from the Baise, Nanning, Guigang, and Wuzhou outbreak isolates lost an IS26 copy and acquired a 5.5 kb region composed of the $intI1$ integron and the $sul1$, $dfrA17$, and $aadA5$ genes (Fig. 4b). Notably, we observed that a small cluster of Chinese cef^R azi^R isolates contained three distinct plasmid replicons. For example, the isolate sh13D178 in this cluster contained the plasmid psh13D178-1 belonging to the IncB/O/K/Z backbone with the $mphA$, $ermB$, $dfrA17$, $aadA5$, and $sul1$ genes; the plasmid psh13D178-2, with the IncFII backbone and harboring the $mphA$, $bla_{TEM-1}$, $aac(3)-IId$, $dfrA17$, $aadA5$, and $sul1$ genes; and the plasmid psh13D178-3, with the backbone IncI1-(Alpha) and containing the $bla_{CMY-2}$ gene (Fig. 4b). The Nanning outbreak strains with col^R acquired an $mcr-1$-positive plasmid of 63 kb in size. The $mcr-1$-positive plasmids carried an IncI2 plasmid backbone similar to that of the first reported $mcr-1$-carrying plasmid pHNSHP45 (GenBank accession no. KP347127 [https://www. ncbi.nlm.nih.gov/nuccore/KP347127.1/])[17]. Compared with the plasmid pHNSHP45, the $mcr-1$-harboring plasmid in the outbreak strains contained an intact IS1×2 mobile element instead of ISApl1 upstream of the $mcr-1$ gene (Fig. 4b). Additionally, we identified an IncI1-(Alpha) plasmid, pSH11sh418-1, containing the $bla_{CTX-M-14}$, $bla_{TEM-1}$, and $aac(3)-IId$ genes, which was present in some Chinese strains with MDR (Fig. 4).

### SNPs and genes associated with cef^R azi^R isolates and outbreak strains

A genome-wide association study (GWAS) was performed to identify whole-genome SNPs that were specific to the cef^R azi^R isolates and outbreak isolates. A total of 59 significant ($P < 1e\text{-}50$) SNPs were found to be associated with outbreak isolates (Fig. 5a), and 60 significant SNPs were associated with cef^R azi^R isolates (Fig. 5b). Ten of these SNPs

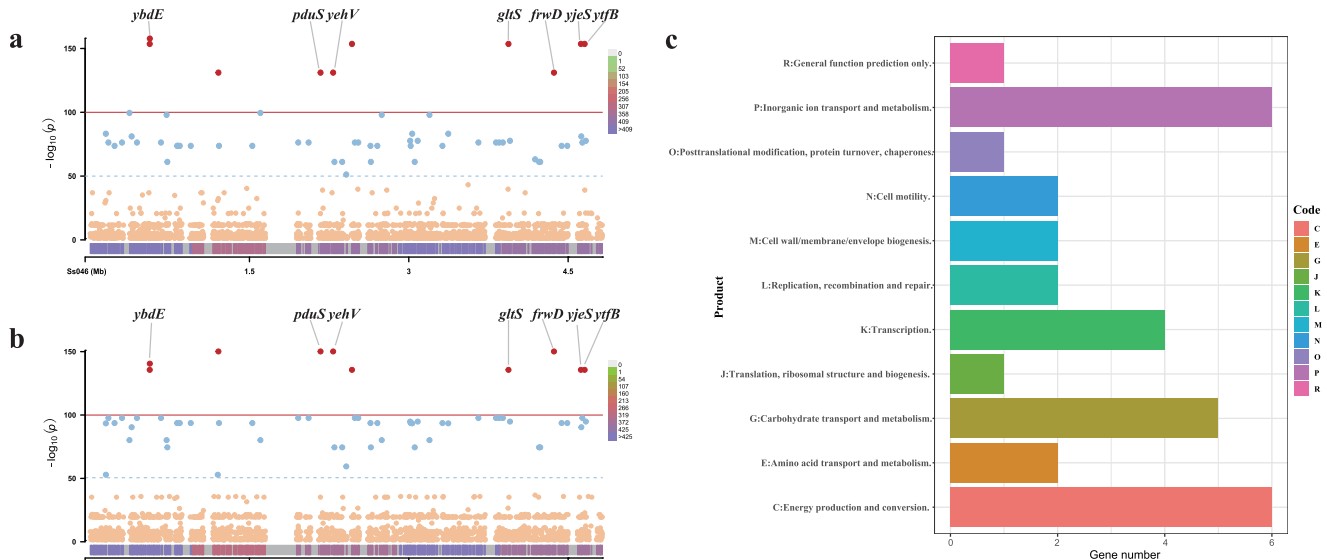

**Fig. 5 | The SNPs identified by a GWAS are associated with cef$^R$azi$^R$ isolates and the outbreak strains. a** and **b** separately show SNPs identified by a GWAS associated with outbreak strains and cef$^R$azi$^R$ strains. The red dots represent significantly associated ($P < $1e-100) SNPs, while the blue dots represent SNPs with a $P$ value <1e-50. The thresholds $P = $1e-100 and $P = $1e-50 are indicated by a red line and a blue line, respectively. The bottom band shows the density of SNP distribution. **c** shows COG function classifications of SNPs with $P < $1e-50. These SNPs are mainly associated with transport, energy production, and transcription. Source data are provided as a Source Data file. SNP single-nucleotide polymorphism, GWAS genome-wide association study, cef$^R$azi$^R$ coresistance to ceftriaxone and azithromycin, COG Cluster of Orthologous Groups.

were identified as being highly significantly ($P < $1e-100) associated with both cef$^R$azi$^R$ isolates and outbreak isolates (Fig. 5 and Supplementary Table S5). In addition to the ten highly significant SNPs, other SNPs found in the cef$^R$azi$^R$ isolates were almost the same as those in the outbreak isolates, with only three SNPs different (one same-sense mutation, one in *gcd*, and one in *fhuE*) (Supplementary Table S5). Among the significantly associated SNPs, 56 were found within protein-coding regions, and four were located within intergenic regions (Supplementary Table S5). Of the coding region SNPs, 32 (57.1%) resulted in missense mutations, 23 (41.1%) were synonymous variants, and one was a nonsense mutation. The nonsynonymous SNPs affected 33 genes with various functions (Supplementary Table S5). The results of Clusters of Orthologous Groups (COG) analysis showed that these genes were mainly associated with energy production and conversion, inorganic ion transport and metabolism, carbohydrate transport and metabolism, and transcription (Fig. 5c). Analysis of virulence-encoding genes showed that there were a total of 97 virulence factor genes, including 70 genes involved in the effector delivery system, three genes involved in exoenzyme activity, one involved in regulation, one involved in enterotoxin activity, one involved in motility, nineteen involved in nutritional or metabolic factor activity, and two involved in other functions. The genes of the effector delivery system, especially the T3SS, motility, regulation, and two other functions, were significantly different in distribution among the lineages. These virulence genes were absent in most of the cef$^R$azi$^R$ strains but present in the strains from the Guigang outbreak and some of the Nanning outbreak strains (Supplementary Fig. S5). To further study the SNPs associated with virulence factors, we extracted all 173 SNPs located in virulence-encoding genes and found that these SNPs were located in 31 virulence genes. Among the SNPs, 91 were missense variations, and 82 were synonymous variations. The 91 missense variations affected 18 genes, including 11 nutritional or metabolic genes (*entA*, *entC*, *entD*, *entF*, *fepA*, *fepE*, *fepG*, *iucA*, *iutA*, *sitA*, and *sitD*), five effector delivery system genes (*ipaH_2*, *ipaH_3*, *rhsA*, *rhsB*, and *vgrG*), one exoenzyme gene (*sigA*), and one gene (*msbB*) with other functions (Supplementary Table S6).

Pangenome analysis based on the 462 isolates showed that a total of 15,340 genes were identified, 12,114 of which were assigned to the accessory genome (Fig. 6 and Supplementary Fig. S6). Among the accessory genes, chi-square tests were performed between the cef$^R$azi$^R$ group and non-cef$^R$azi$^R$ strains, and 92 specific genes significantly associated with the cef$^R$azi$^R$ phenotype with a $P$ value below 1e-30 were selected, which was more than 80% of the cef$^R$azi$^R$ group but less than 20% of the non-cef$^R$azi$^R$ isolates (Fig. 6a and Supplementary Table S7). COG analysis showed that 14 genes mapped to nine COG clusters and were mainly involved in transcription, replication, recombination and repair, cell wall/membrane/envelope biogenesis, and defense mechanisms (Fig. 6b and Supplementary Table S7).

## Discussion

Shigellosis outbreaks caused by MDR *S. sonnei* are now being commonly reported[18,19]. Nevertheless, outbreaks caused by MDR *S. sonnei* with cef$^R$azi$^R$ resistance have rarely been reported[14,16]. To date, no outbreaks have been reported as being caused by MDR *S. sonnei* with col$^R$. Concerningly, here, we report the prevalence and outbreaks of shigellosis caused by *S. sonnei* with a combination of MDR, cef$^R$azi$^R$, reduced susceptibility to fluoroquinolones, and even col$^R$. *S. sonnei* clones belonging to Lineage III appear to be more easily transmitted from person to person and spread across regions[12] and to be highly proficient at acquiring resistance to additional antimicrobials, such as fluoroquinolones, macrolides, third-generation cephalosporins, and polymyxins, when they are introduced into new locations[20,21]. Thus, the clone in this study belonging to Lineage III seemingly has the potential to develop into a superbug, posing the risk of nearly untreatable shigellosis.

To our knowledge, this is one of the few studies that has used WGS to explore the microevolution and phylogeography of Chinese *S. sonnei* isolates, especially those with cef$^R$azi$^R$ and even col$^R$. Our data showed that the Chinese *S. sonnei* isolates formed a main Chinese clade within the global Lineage III that may have emerged circa 1987. The cef$^R$azi$^R$ isolates were mainly located in the main Chinese clade, which was probably derived from the non-cef$^R$azi$^R$ *S. sonnei* ancestor that

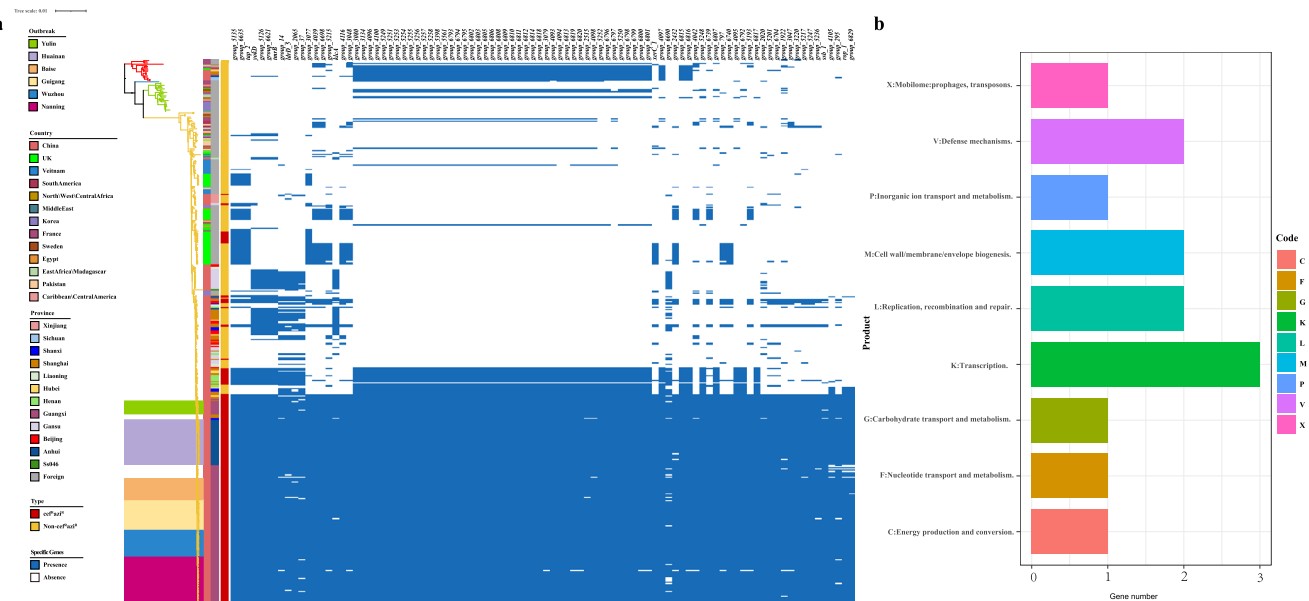

**Fig. 6 | Accessory genes identified by pangenome analysis that might be associated with cef$^R$azi$^R$ isolates and the outbreak strains. a** shows the accessory genes significantly associated with the cef$^R$azi$^R$ phenotype. There were 92 genes screened by the chi-square test whose *P* values were lower than 1e-30 and defined as genes if their proportion was more than 80% in the cef$^R$azi$^R$ group but less than 20%   in non-cef$^R$azi$^R$ isolates. **b** COG function classifications of significant accessory clusters of orthologous genes. Source data are provided as a Source Data file. cef$^R$azi$^R$ coresistance to ceftriaxone and azithromycin, COG Cluster of Orthologous Groups.

existed circa 1999 and then spread and localized into a more recently emerged clone of *S. sonnei* with XDR. This distinct clone has now disseminated to diverse regions of China, possibly with Shanghai acting as a major transport hub for the spread of the cef$^R$azi$^R$ *S. sonnei* clone. Shanghai, an international metropolis, is the largest economic and trade center of China, with a resident population of 24 million and a floating population of -10 million, possibly facilitating the spread of this specific clone. These analyses indicated that Shanghai may act as a hub for the spread of the cef$^R$azi$^R$ *S. sonnei* in China, and this clone has spread successfully within diverse regions of China as a single, rapidly evolving clone, establishing new, local populations and causing large-scale shigellosis outbreaks.

The severe MDR of this specific clone may be due to the strong selective pressures imposed by antibiotic usage. Antibiotic usage in China is complex and widespread, and China has become the largest producer and consumer of antibiotics in the world[22]. Cephalosporins, macrolides, fluoroquinolones, and penicillins are among the most commonly used antibiotics in China[23,24], and colistin, one of the most commonly used antibiotics worldwide, is used in agriculture in China[17]. Fluoroquinolones, cephalosporins, and macrolides are recommended first-line drugs for shigellosis treatment[25,26]. Combinations of azithromycin and colistin have been found to represent a potentially invaluable option for the treatment of infections caused by MDR Gram-negative bacteria[27]. However, colistin has recently been banned from use in animal feeds but approved for use in human beings in China, and this change will increase colistin resistance rates in clinical *Enterobacteriaceae* isolates[28,29]. Moreover, we recently reported a new and serious antibiotic crisis in China, revealing antibiotic pollution in the food and drinking water supply[30]. Therefore, the heavy use of antibiotics such as cephalosporins, macrolides, fluoroquinolones, and colistin could have resulted in sufficiently high and sustained selective pressure in the clinical, veterinary, and environmental settings, leading to the emergence, spread, and maintenance of the severe MDR *S. sonnei* strains observed in this study.

AMR phenotypes are conferred largely by either chromosomal gene mutations or the acquisition of new AMR determinants through horizontal gene transfer (HGT). Acquisition of AMR determinants

frequently precedes the spread and clonal expansion of bacterial pathogens, for example, the *S. sonnei* global Lineage III with ciprofloxacin and azithromycin resistance and the azithromycin-resistant *S. flexneri* 3a lineage circulating in the MSM community[9,12]. Here, we demonstrated that the emergence of XDR, especially the cef$^R$azi$^R$ and col$^R$ phenotypes, was attributed to the presence of AMR plasmids, which can spread easily between *Shigella* and other species of *Enterobacteriaceae*[31]. We observed that a UK clone with cef$^R$azi$^R$ had an IncFII plasmid, p183660, harboring the *bla*$_{CTX-M-27}$, *bla*$_{TEM-1}$, *mphA*, *ermB*, *dfrA17*, *aadA5*, and *sul1* genes. This plasmid was identified from *S. sonnei* isolates with the cef$^R$azi$^R$ phenotype from MSM in the UK[16], exhibiting >95% identity to pKSR100 (GenBank accession no. LN624486 [https://www.ncbi.nlm.nih.gov/nuccore/LN624486.1/]), an MDR plasmid from an *S. flexneri* 3a outbreak associated with infections among MSM in the UK[32]. Notably, the cef$^R$azi$^R$ phenotype for the Chinese isolates, including all of the outbreak strains, was mainly attributed to the presence of a distinct 109 kbp plasmid belonging to the IncB/O/K/Z incompatibility group, coharboring the *bla*$_{CTX-M-14}$, *mphA*, *aac(3)-IId*, *dfrA17*, *aadA5*, and *sul1* genes. This plasmid was genetically close to the plasmid pSH15sh99[14], which was present in the Yulin and Huainan outbreak strains. The cef$^R$azi$^R$ clone with the pSH15sh99-like plasmid spread in Guangxi, Shanghai, Anhui, Hubei, Shanxi, and Henan Provinces and caused six shigellosis outbreaks in Guangxi and Anhui. These results indicated that this distinct clone harboring the *bla*$_{CTX-M-14}$- and *mphA*-positive plasmids might have been prevalent in the local populations and spread interregionally.

Additionally, the cef$^R$azi$^R$ phenotype from a small cluster of Chinese cef$^R$azi$^R$ isolates identified from Shanghai, Hubei, and Henan Provinces was attributed to the presence of three different plasmids: one was the pSH15sh99-like IncB/O/K/Z plasmid psh13D178-1 harboring the *mphA*, *ermB*, *dfrA17*, *aadA5*, and *sul1* genes, one was the pKSR100-like IncFII plasmid psh13D178-2 with the *mphA*, *bla*$_{TEM-1}$, *aac(3)-IId*, *dfrA17*, *aadA5*, and *sul1* genes, and one was the IncI1-(Alpha) plasmid psh13D178-3 containing the *bla*$_{CMY-2}$ gene, which was similar to the plasmid pV233-b (GenBank accession no. LC056425 [https://www.ncbi.nlm.nih.gov/nuccore/LC056425.1/]) detected in MDR *Escherichia coli* isolates from water samples in India[33]. Among the

colistin-resistant outbreak strains, the *mcr-1* gene was located on a plasmid with close homology to pHNSHP45, which was originally identified in clinical and veterinary *E. coli* and *K. pneumoniae* isolates[17] and is now recognized as one of the most prevalent *mcr-1*-harboring plasmids among *Enterobacteriaceae* in diverse sources, including foods, animals, environmental samples, patients and healthy persons from China[29], suggesting a potential spread of this plasmid type between diverse bacterial species. The *mcr-1* gene has rarely been reported in *Shigella* species. We have reported the emergence of *mcr-1*-positive *S. sonnei* strains in China[34], and here, we report the first outbreak of shigellosis caused by *mcr-1*-positive *S. sonnei* strains with the cef^R^azi^R^ phenotype. The above analyses indicated that the cef^R^azi^R^ and col^R^ phenotypes were attributed to the acquisition of diverse distinct plasmids harboring the $bla_{CTX-M-14}$, $bla_{CMY-2}$, $bla_{TEM-1}$, *mphA/ermB*, and *mcr-1* genes, and these distinct plasmids might have been prevalent in the local populations and spread interregionally and seem to be evolving toward local establishment. Concerningly, because *S. sonnei* is a human-adapted pathogen with no known reservoir outside the human population, there is the potential for these XDR strains to serve as a vehicle for the rapid propagation of AMR genes among human-associated *Enterobacteriaceae* more widely, which would represent a major threat to global public health and thus have much broader implications.

The COG results of phenotype-related pangenomic genes and SNP sites showed that all these genes involved were related to energy production; carbohydrate transport and metabolism; transcription, replication, recombination, and repair; cell wall or membrane or envelope biogenesis, and other functions. Although the effects of these related genes on the biological functions of the strains cannot be determined by simple mathematical addition of the number of genes involved in these two COGs, there were many more genes associated with transport and energy production, such as the *fep* gene encoding iron enterobactin, *msb* gene encoding a full transporter, and *tauB* gene encoding the taurine ATP-binding component of a transport system[35,36]. We hypothesize that the strains with the cef^R^azi^R^ phenotype may have stronger nutrient transport and metabolic capacity, and to some extent, this may contribute to better adaptation to the environment of these strains. In addition, cell wall and membrane and envelope biogenesis[37]-associated genes, such as the phage-related lysozyme gene *rrrD*, may play a role in adaptation to poor environments. These may be important reasons why certain waterborne strains can survive in water and contribute to a series of outbreaks. This provides strong evidence of the divergent evolution of *S. sonnei* by the accumulation of genomic signatures, which may be associated with the local establishment and niche adaptation of these cef^R^azi^R^ isolates, especially the XDR outbreak clone.

Factors other than AMR and genetic elements, especially climate changes such as precipitations and floods, are of particular relevance to the transmission of infectious diseases, particularly waterborne and foodborne diseases. Studies have shown that a significant increase was observed in bacillary dysentery infectious diarrhea and in other infectious diarrhea after floods[38,39]. Guangxi, with the Pearl River meandering through it (Supplementary Fig. S2) and Anhui, with the Yangtze River and Huai River flowing through it (Supplementary Fig. S2), are among the provinces that have experienced serious flood disasters in China[38,39]. Meteorological data from the National Bureau of Statistics of China demonstrate that the strong 2015-2016 El Niño event caused a serious flood disaster in 2016 in southern China, especially in the basins of the Yangtze River and Pearl River (Supplementary Fig. S2)[40]. The outbreaks in this study mainly occurred in kindergartens and schools in rural and urban areas of China, which had crowded populations, compromised hygiene practices, and poor access to safe water and sanitation. During and after heavy precipitation and floods, living environments such as temperature, humidity, and surface vegetation might change, water supply systems are damaged, and

sewage systems and waste-disposal systems are destroyed, leading to the contamination of drinking water facilities and to a more susceptible population and providing a suitable environment for the mobilization and spread of infectious diarrheal pathogens. These changes may have increased the transmission of enteric pathogens in flood-affected areas and promoted a quick interregional spread through contaminated water. According to the above analyses, combined with conventional epidemiological and genomic epidemiological investigations, we obtained strong evidence that heavy precipitations and floods can increase the risk of shigellosis infections and outbreaks and reported vivid examples of waterborne outbreaks of shigellosis caused by MDR *S. sonnei* with the cef^R^azi^R^ and/or col^R^ phenotype associated with water contamination due to heavy precipitations and floods.

In conclusion, we report an XDR *S. sonnei* clone with cef^R^azi^R^, reduced susceptibility to fluoroquinolones and even col^R^, associated with a series of waterborne outbreaks of shigellosis in China. The emergence and prevalence of this most recently emergent XDR clone may be driven by antibiotic usage and climate change, such as heavy precipitation or floods, and by horizontally acquired AMR determinants, including an *mcr-1*-harboring plasmid and a $bla_{CTX-M-14}$- and *mphA*-encoding plasmid. Coupled with globalization and the increased movement of people across national borders, this particular clone has the potential to spread globally, posing a risk of the spread of highly resistant shigellosis. Continued surveillance and further genomic epidemiological studies are urgently required to determine the potential dissemination and consequent risk to global public health to inform the formulation of public health policies, including vaccine strategies, rational use of antimicrobials, conventional species identification, and international source tracking.

## Methods

### Ethical approval

The study was approved by the review board of the Chinese PLA CDC, Beijing, China (No. #20220523). All patients provided informed consent for the collection of samples and relevant information.

### Study design

In 2015, we reported a waterborne outbreak (Yulin outbreak) of shigellosis caused by MDR *S. sonnei* with cef^R^azi^R^ in a kindergarten in Yulin City, Guangxi Zhuang Autonomous Region of China[14]. Thereafter, five additional outbreaks of shigellosis caused by MDR *S. sonnei* with cef^R^azi^R^ were identified, including four in Guangxi and one in Anhui Province (Supplementary Table S1 and Supplementary Fig. S7). One (Baise outbreak) occurred in a primary school in Baise city, with 93 reported cases from June 28 to July 10 in 2016, and another (Nanning outbreak) occurred in a middle school in Nanning city, with 120 reported cases from September 22 to October 2 in 2016. The third (Guigang outbreak) occurred in a kindergarten in Guigang city, with 52 reported cases from April 17 to April 22 in 2019, and the fourth (Wuzhou outbreak) occurred in a middle school in Wuzhou city, with 22 reported cases from September 6 to September 9 in 2019. The remaining (Huainan outbreak) occurred in a kindergarten and a primary school in Huainan City, Anhui Province, China, with 598 reported cases from August 17 to August 28 in 2020.

This alarming trend prompted us to investigate the six outbreaks and explore the link between the outbreak and the historical strains recovered from our routine surveillance of shigellosis. Field investigations were conducted, and demographic and epidemiologic data were collected from these outbreaks by a standardized survey. The patients were sent to the local hospital for quarantine, review, and treatment by physicians. Clinical cases were defined as persons with two or more of the following symptoms of acute gastroenteritis during the outbreak period: diarrhea (more than three times per day), stool abnormalities (onset of watery, mucous, or bloody purulent stool), nausea, and/or vomiting, abdominal pain and fever (≥38 °C).

Laboratory-confirmed cases were defined as patients with positive results for *S. sonnei* detection by bacterial isolation or stool DNA-based real-time polymerase chain reaction (RT–PCR). The details of food production, the water and sewage system, and food hygiene policies and procedures were reviewed. The meteorological data, especially the annual precipitation and flood disaster data collected from the National Bureau of Statistics of China (http://www.stats.gov.cn/), were also considered. All food and water samples were collected by sterile sampling according to the national food safety standards[41], such as GB 4789.1-2016, GB 4789.2-2016, GB 4789.3-2016, GB 4789.4-2016, GB 4789.6-2016, and GB 4789.10-2016, and the standards for drinking water quality (GB 5749-2006)[42]. The samples were screened for *Shigella* spp. using RT–PCR assays (Shanghai BioGerm Medical Biotechnology Co., Ltd., Shanghai, China), and suspected pathogenic bacteria were isolated through standard procedures by local CDCs. Water samples were filtered through a 0.45 μm sterile membrane (Millipore, USA) using a vacuum filtration apparatus and then inoculated on xylose lysine deoxycholate (XLD) agar plates, *Salmonella-Shigella* agar (SS agar), and/or *Salmonella* chromogenic agar plates (Beijing Land Bridge Technology Co., Ltd., China). Suspected isolates recovered from the samples were identified using a commercial biochemical test kit (API 20E system; bioMérieux Vitek, France) according to the manufacturer's recommendations.

### Microbiological investigation

The outbreak and sporadic *S. sonnei* isolates in this study were identified by API 20E biochemical tests (bioMerieux, Marcy l'Etoile, France) and serotyping with a commercial antiserum kit (Denka Seiken, Tokyo, Japan). The minimal inhibitory concentrations (MICs) of colistin and azithromycin were determined by the broth microdilution method with concentrations of colistin ranging from 0.5 to 32 μg/mL and with concentrations of azithromycin ranging from 4 to 256 μg/mL. Susceptibility to other antimicrobials was determined by broth microdilution using a 96-well microtiter plate (Sensititre, Trek Diagnostic Systems, Thermo Fisher Scientific Inc.) according to the manufacturer's instructions, and MICs were interpreted according to the CLSI guidelines[43]. In this study, MDR strains were defined as those with resistance to three or more categories of antibiotics, and XDR was defined as MDR with additional resistance to azithromycin, ceftriaxone, or colistin. The presence of $bla_{CTX-M}$, *mphA*, and *mcr-1* genes was screened by PCR and Sanger sequencing using the primers designed in this study, which were also used in the previous studies[14,17,44]. To determine the location of the $bla_{CTX-M}$, *mphA*, and *mcr-1* genes, S1–PFGE and Southern blotting were performed. The primer and probe sequences used in this study (Supplementary Table S8) were synthesized by Sangon Biotech (Shanghai) Co., Ltd.

### Whole-genome sequencing

In this study, we performed WGS of a total of 307 Chinese *S. sonnei* isolates recovered from 2004 to 2020, including 155 isolates from the six outbreaks and 152 sporadic isolates covering diverse regions of China (Fig. 1a and Supplementary Table S3). Genomic DNA was extracted from the Chinese strains by the QIAamp DNA Mini Kit (Qiagen, Hilden, Germany). WGS was performed by Illumina HiSeq and MiSeq next-generation sequencing platforms as instructed by the manufacturer (Illumina, San Diego, CA, USA). The Illumina base-calling pipelines were used to process the raw fluorescent images and call sequences. The quality of the sequences was verified using Trimmomatic v0.39[45]. After the removal of adapter sequences and low-quality paired-end reads, the remaining high-quality reads were de novo assembled using Spades v3.15.2[46]. Additionally, we representatively selected five $bla_{CTX-M}$-, $bla_{CMY-2}$-, *mphA*-, and *mcr-1*-harboring strains for long-read genome sequencing with an Oxford Nanopore MinION sequencer. De novo assemblies were generated using Canu v1.6[47], and

errors of sequencing were corrected by the Illumina sequencing data using Pilon v1.24[48]. All the sequencing data have been submitted to NCBI Sequence Read Archive under the BioProject number PRJNA835603 [https://www.ncbi.nlm.nih.gov/biosample/?LinkName=bioproject_biosample_all&from_uid=835603].

### Phylogenetic analysis

Reads from each isolate and those from previously published genomes were mapped to the reference genome *S. sonnei* Ss046 (GenBank accession no. NC_007384 [https://www.ncbi.nlm.nih.gov/nuccore/NC_007384.1/]) to identify SNPs and the final core SNPs using the RedDog v1beta.10.3 pipeline (https://github.com/katholt/reddog). Briefly, RedDog uses Bowtie2[49] for read mapping and SamTools[50] for SNP calling. Filtering of low-quality SNP calls and horizontally transferred regions was performed—SNPs with more than 5% missing alleles were filtered out[3,7]. A maximum-likelihood phylogenetic tree was performed to explore the population structure by FastTree v2.1.8[51]. To infer the evolutionary dynamics of each branch, we selected 209 representative strains with definite isolation times and regions and used Bayesian evolutionary analysis sampling trees (BEAST2 v2.6.6) to reconstruct the tree. A Bayesian skyline model with a log-normally distributed clock rate was chosen, and Markov chain Monte Carlo (MCMC) generations were set to 300 million steps with samples taken every 5000 MCMC generations in 6 independent MCMC chains. Finally, we burned in the first 10% steps of each chain and combined the log files of these chains before building the final maximum clade credibility (MCC) tree. Figtree was used to display the MCC tree produced by BEAST. The beautification of the phylogenetic tree and the visualization of associated data were carried out on the iTOL website (https://itol.embl.de/). Further genotyping of the strains was performed using Mykrobe v0.12.1 according to the study of ref. 13. MLST analysis was conducted using the MLST tool v2.19.0[52] according to the database of the PubMLST website[53].

### Identification of mobile genetic elements

The AMR genes were detected using Resistance Gene Identifier (RGI) v5.1.1, referring to the Comprehensive Antibiotic Resistance Database (CARD)[54]. To determine the $bla_{CTX-M-14}$-, *mphA*-, and *mcr-1*-harboring plasmid sequences, we extracted fragments of plasmids using a Qiagen Plasmid Midi kit (Qiagen, Germany) and submitted them for sequencing using the Illumina MiSeq platform to ensure that most of the fragments sequenced belonged to plasmids. We performed quality control using Trimmomatic and assembly using plasmidSPAdes[55] on the plasmid sequencing data before mapping them to the Plasmidfinder database[56]. The complete plasmid sequences were annotated using Prokka v1.14.6[57], and the annotated plasmid sequences have been deposited in GenBank under accession numbers MG299127-MG299153 and ON461899-ON461902. The locations of plasmid replicons, IS sequences and AMR genes were determined using PlasmidFinder v2.0.1[56], ISfinder[58], and ResFinder v4.0.1[59], respectively. Plasmid sequence comparison and map generation were performed using BLAST v2.12.0[60] and custom Perl scripts, respectively. Plasmids of four types of replicons (pSH15sh99, pH267-2, and pSH13D178-1 of IncB/O/K/Z; pKSR100, p183660, and pSH13D178-2 of IncFII; pV233-b, pSH13D178-3, and pSH11sh418-1 of IncI1-(Alpha); and pHNSHP45 and pSH267-1 of IncI2) containing the *mcr-1*, $bla_{CTX-M}$, $bla_{CMY-2}$, $bla_{TEM}$, and *mphA* genes were selected for further analysis. Genomic assemblies were mapped against the selected plasmids using BLAST to determine whether the strains carried the corresponding plasmids, the assembly sequences covered over 90% of the length of plasmids, and the identity between query sequences and subject sequences exceeded 70%. In addition, if the assembly sequences of the strains included AMR genes that were harbored in corresponding plasmids, we concluded that the

strains carried the same plasmids or plasmids with a similar structure. The WGS data and representative plasmids were mapped to the local INTEGRALL[61] database to identify the type of integron using BLAST.

### Analysis of potential markers and related genes

Based on annotated assemblies in gff format produced by Prokka[57], a pangenome analysis was performed for all 462 isolates to clarify the distribution of pangenes in these strains using Roary v3.13.0[62]. Genes were defined as cef$^R$azi$^R$-specific genes if their proportion was more than 80% in the cef$^R$azi$^R$ group but less than 20% in other isolates. A GWAS was also performed to identify significant SNPs associated with the cef$^R$azi$^R$ phenotype and outbreak strains. The Snippy pipeline was used to call the SNPs in all the assemblies, and strains were spread into the cef$^R$azi$^R$ group and other groups according to the presence and absence of the *mphA* and *bla*$_{CTX-M}$ or *bla*$_{cmy}$ genes. A GWAS was performed using the Plink v1.07[63] pipeline to calculate the *P* value of each SNP, and *P* values below 1e-100 were considered significant. The *P* value and locus in reference Ss046 of SNPs with *P* values lower than 0.05 are shown in a Manhattan plot. The SNPs above the red line were significant. Additionally, fluoroquinolone resistance-associated mutations in the *gyrA* gene were extracted from the genome-wide SNP calls[3]. COGs of genes identified by pangenome analysis and a GWAS were determined by mapping to the COG function database using BLAST and visualized using R v4.02. Considering that virulence genes may play certain roles, we also mapped WGS data to the VFDB[64] using BLAST to identify the virulence-encoding genes and displayed the results in a heatmap.

### Statistical analysis

Between-group comparisons were performed using Chi-square tests. Quantitative statistics of drug resistance genes were completed using R. AMR gene numbers of the strains carried in different groups and changes in the rainfall and flood area were also visualized using R. A Kruskal–Wallis test for resistance genes carrying was performed between lineages, while Wilcoxon tests were performed between cef$^R$azi$^R$ strains and non-cef$^R$azi$^R$ strains, cef$^R$azi$^R$ strains and non-cef$^R$azi$^R$ strains in Lineage III, outbreak strains and sporadic strains in the cef$^R$azi$^R$ group. Other descriptive data were summarized according to descriptive statistics. *P* values less than 0.05 were considered to indicate statistical significance.

### Reporting summary

Further information on research design is available in the Nature Portfolio Reporting Summary linked to this article.

## Data availability

The publicly available sequences used in this study are available in GenBank under accession numbers KX008967, KY471628, KP347127, LN624486, LC056425, and NC_007384. The sequencing data generated in this study have been deposited in NCBI Sequence Read Archive under the BioProject number PRJNA835603. The annotated plasmid sequences in this study have been deposited in GenBank under accession numbers MG299127-MG299153 and ON461899-ON461902. Source data are provided with this paper.

## Code availability

The code that supports the findings of this study is available from the corresponding authors on request. Code requestors will be required to sign a code access agreement in accordance with the confidentiality requirements of the author's affiliations.

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

## Acknowledgements

This work was funded by the National Key R&D Program of China (grant number 2017YFC1600100 to S.Q.), the National Key Program for Infectious Diseases of China (no. 2012ZX10004215 to S.Q.), the National Nature Science Foundation of China (nos. 82173580 and 81872678 to S.Q.), and the Chinese Academy of Medical Sciences (CAMS) Innovation

Fund for Medical Sciences (2021-I2M-1-044 to J.Y.). The funders had no role in the study design, data collection, data analysis, data interpretation, or writing of the report.

## Author contributions

S.Q., K.E.H., and H.S. designed the study. S.Q., X.X., M.M., J.Z., S.L., D.Y., H.S., Y.S., and J.Y. collected the data. S.Q., J.H., C.Y., X.D., H.L., L.W., K.L., K.Z., M.Y., Y.X., and X.D. analyzed and interpreted the data. S.Q., J.H., K.E.H., H.S., H.Z., K.L., S.B., Y.X., and K.M. wrote the report. All authors reviewed, revised, and approved the final report.

## Competing interests

The authors declare no competing interests.
