## [Peer review file · Nature Communications]

REVIEWER COMMENTS

Reviewer #1 (Remarks to the Author):

This work describes about the molecular characteristics of a an extensively drug-resistant clone of *Shigella sonnei* associated with waterborne outbreaks of shigellosis in China. This work has covered susceptibility patterns of the pathogen, plasmid analysis, antimicrobial resistance genes and phylogenetic analysis using the whole genome sequence.

Comments.

1. Abstract is not reflecting the all the findings made in this investigation.
2. What is the source of the water, pipeline/stored water? Evidence of waterborne outbreaks is very weak. Why only children are the affected group in all these six outbreaks
3. Does these outbreaks repeatedly appeared in the same geographical location (more specifically the same school) during 2015-2020?
4. Introduction: explain the basics of the lineages (Hot et al., 2012) instead of abruptly citing the Lineage III).
5. Line 112. Mention the period and source of isolation of these 152 sporadic strains from historical surveillance of shigellosis.
6. What is the rational for using PFGE in this study? The plasmids would have been extracted directly from the isolates and tested for the AMR genes. The motioned 60-100 kp fragment might contained chromosomal DNA fragment also. Of course, the Inc nature of the plasmids gives indirect evidence.
7. Lines 170-172. Restructure the sentence. The AMR genes are more than the corresponding antimicrobials with the usage of the word "respectively".

8. Line 200-201. The 5.5 kb region with *sul1*, *dfrA17*, and *aadA5* might be associated with an integron.

9. Most of the AMR genes mentioned in lines 170-172, 205 might have located in the integrons. It is worth to identify the type of integrons, mostly the *Int1* and *Int2* types. This information may also be included in the phylogenetic analysis.

10. Line 413. Mention stool DNA-based real-time polymerase chain reaction (RT-PCR) detection

11. Most importantly, the virulence encoding genes are missing in the analysis. Epidemiologically, it is important to undertake this kind of analysis, as it is directly associated with the diseases. Authors can generate this formation using the WGS data.

12. The SNPs analysis has to be extended in virulence encoding genes and its transcriptions and functions.

13. It is also suggested that the WGS data can be explored for MLST/MLVA/VNTR analysis to identify the sequence types.

14. Discussion section mostly reflects the information provided in the results and this can be avoided.

15. Methodology for the isolation of shigellae from water/food samples was not mentioned.

16. Reference numbers mentioned in the text and in the concerned section are not matching.

17. Finally, the English language has to be improved. In many places, grammar and syntax errors exists.

Reviewer #2 (Remarks to the Author):

This study reports on the sampling and genomic epidemiology investigation of a large sample of outbreak- and sporadic isolates of *S. sonnei* collected in China. Applied methods are scientifically sound. The Chinese isolates are analyzed in context of a global *S. sonnei* strain panel and the previously established lineages to assess the genome plasticity and differentiating features. The in-depth analysis discusses chromosome- and plasmid content and inventories along with the respective XDR profiles. Overlaying meteorological data is a strength. The discussion is to some extent redundant and reiterates the results e.g. SNP, COG, and plasmids compositions in granular detail; it should be more concise to highlight and interpret key findings. The two COG categorizations are somewhat superficial compared the level of detail presented elsewhere and is not particularly insightful. These data should maybe be discussed in context to what is observed in the other lineages/other outbreaks. Use of the “first time/study” seems a bit overstated throughout the MS given the novelty of original research. Some methods need clarification. Overall this work will contribute to the global understanding of shigella pathogenome evolution and the AR virulence traits of emerging clones that pose public health threats.

Line 41: report (I assume the statement refers to the current study with the additional AR profile elucidated and not the previous work - citation 13)

Line 91: a plasmid

Line 115: the cefR

Line 135: The majority of Chinese

Line 138: demonstrating > suggesting

Line 141: may have emerged

Line 155: Were these relationships expected based on geography/flooding etc. (Fig 1)

Line 179: Why is the acquisition of yet another resistance surprising?

Line 185: phenotypes

Line 189: rephrase

Line 203: the backbone

Line 205: the backbone

Line 222: elaborate on “almost the same”

Line 243: have been rarely

Line 245: report

Line: 257: may have emerged

Line 307: locally, in the local population

Line 322: rephrase ever reported

Line 328: in the process > towards?

Line 333: Suggest to delete > ... than just the treatment of Shigellosis.

Line 336: While... ? check intended meaning of this sentence.

Line 380: globalization

Line 404: prompted

Line 405: outbreak and historical strains

Line 417: are presented > were considered?

Line 452: errors of sequence were corrected ... Which tool was used for error correction?

Line 476: Plasmid sequence data were performed for quality control ...

Please rephrase.

Line 482: Plasmid sequence comparison and map generation were performed using BLAST, respectively.

Which tool was used to generate the linear plasmid displays?

Line 507: analyses

Reviewer #3 (Remarks to the Author):

The manuscript titled “An extensively drug-resistant clone of *Shigella sonnei* with resistance to azithromycin, ceftriaxone, and/or colistin associated with waterborne outbreaks of shigellosis in China” used whole-genome sequencing to describe XDR *S. sonnei* from outbreaks of diarrhea in China. The authors compared isolates from six shigellosis outbreaks in China to sporadic isolates from China as well as previously described global *S. sonnei*, providing important insight into the expansion of this XDR clone in China.

XDR and MDR: How were the isolates determined to be XDR and MDR? How many antimicrobial categories did the isolates exhibit resistance to? Please define in the introduction and/or methods.

Line 41: Change “reported” to “report”. Also, it is not clear what is meant by a distinct phylogenetic group. How does the XDR clone fit within the genotyping scheme described by Hawkey et al. 2021 (PMID: 33976138)?

Line 45: What is the number of outbreaks?

Line 48: Change “with *mcr-1* gene” to “with an *mcr-1* gene”.

Line 72: Please add a reference.

Line 74: Change “locates” to “located” and “Global III Lineage” to “global Lineage III”.

Lines 77-78: Change to “has been increasingly detected among MDR *S. sonnei*”

Line 82: Change “within” to “at”.

Lines 84-85: Please elaborate on why these groups have great public health significance.

Lines 90-92: Are these recent outbreaks first described in this study? Or can references be provided where these have been described in other publications?

Line 93: Should this be “with concurrent resistance to”?

Lines 95-96: Please provide references.

Line 104: Additional details of the outbreaks such as age of the patients and years would be informative to include here. I would suggest moving the first two sentences from the supplementary results to this section.

Line 141: Should this say “may have emerged”?

Line 146: Is this referencing the supplementary data table? Or the analyses described in the supplementary results? Please clarify.

Line 169: Should these be Figures S1A and S3?

Lines 186-187: Should this be “four plasmids representing different replicons”?

Line 202: Should this be “three distinct plasmid replicons”?

Line 217: For consistency the $-\log_{10}(P)$ should be represented as p-value as written later in this section.

Lines 236-239: How are these mutations potentially significant to expansion of the MDR clones? This can be elaborated upon in the discussion (see comment, Lines 335-345).

Lines 247-251: Is this description also based on published findings? Please provide the references.

Line 257: Should this be “may have emerged”?

Line 271: Change “antibiotics” to “antibiotic”.

Line 307: Should this be “might have been prevalent locally”? Or in “local populations”?

Line 322: It is not clear what is meant by “We had ever reported”. Please clarify.

Lines 335-345: These lines are repetitious with what is in the results and much of this detail could be removed to help shorten the discussion. Provide more interpretation of how mutations in these types of genes may facilitate the expansion of the MDR isolates.

Lines 374-375: Remove MDR as XDR are by definition MDR. It would be informative here to describe the number of antibiotic categories to which the XDR clone exhibits resistance.

Line 382: Does this mean the potential to spread globally?

Lines 490-492: This sentence is not clear. Please revise.

Line 529: Were the genome assemblies and sequence reads of each isolate analyzed by WGS in this study deposited into a public repository such as GenBank? This should be described here and accession numbers of assemblies and SRAs should be added to the isolate table.

Fig S1: How were the ladder images generated that are overlayed on the southern? Were the ladders from an inverse exposure of panel A? In the figure legend please mention the ladder used and describe that panels B, C, and D contain the ladders from the ethidium-stained gel prior to transfer, overlayed on the southern hybridization images. Also, please add the ladder band sizes to the bottom row of images.

Extended Data Figure 1 and Extended Data Table 1: It is confusing to have a supplemental Figure S1, and also an Extended Data Figure 1. Can the extended data table and figure be included among the supplementary tables and figures?

Responses to Reviewers' Comments

We thank the reviewers for the valuable and constructive comments. We have carefully incorporated the reviewers' advice and, as a result, our paper has improved substantially.

Below, we explain in detail how we have taken each of the comments into account in the revision of our manuscript. We show the reviewer's comments in italics and reply to them in the standard font. All the revisions have been highlighted in the revised manuscript.

Responses to Reviewer #1

This work describes about the molecular characteristics of a an extensively drug-resistant clone of Shigella sonnei associated with waterborne outbreaks of shigellosis in China. This work has covered susceptibility patterns of the pathogen, plasmid analysis, antimicrobial resistance genes and phylogenetic analysis using the whole genome sequence.

Comments:

- 1. Abstract is not reflecting the all the findings made in this investigation.*

Response: Thank you for pointing out this problem. Since *Nature Communications* requires the Abstract section to be no longer than 150 words, we omitted some secondary findings in the original manuscript. In the revised manuscript, in response to your comment, we have rewritten the Abstract section to summarize our findings as comprehensively as possible:

“Antimicrobial resistance of *Shigella sonnei* has become a global concern. Here, we report a phylogenetic group of *S. sonnei* with extensive drug resistance, including a combination of multidrug resistance, coresistance to ceftriaxone and azithromycin

(cef^Razi^R), reduced susceptibility to fluoroquinolones, and even colistin resistance (col^R). This distinct clone caused six waterborne shigellosis outbreaks in China from 2015 to 2020. We collected 155 outbreak isolates and 152 sporadic isolates. The cef^Razi^R isolates, including outbreak strains, were mainly distributed in a distinct clade located in global Lineage III. The outbreak strains formed a recently derived monophyletic group that may have emerged circa 2010. The cef^Razi^R and col^R phenotypes were attributed to the acquisition of different plasmids, particularly the IncB/O/K/Z plasmid coharboring the *bla*_{CTX-M-14}, *mphA*, *aac(3)-IIId*, *dfrA17*, *aadA5*, and *sulI* genes and the IncI2 plasmid with an *mcr-1* gene. Genetic analyses identified 92 specific accessory genes and 60 single-nucleotide polymorphisms associated with the cef^Razi^R phenotype. Surveillance of this clone is required to determine its dissemination and threat to global public health.” (Lines 39-52)

2. *What is the source of the water, pipeline/stored water? Evidence of waterborne outbreaks is very weak. Why only children are the affected group in all these six outbreaks*

Response: Thanks for your comment. In this study, investigations on water supply systems were performed for five of the six outbreaks (all except for the Wuzhou outbreak). For the locations of the Yulin outbreak, Nanning outbreak, and Guigang outbreak, water supplies were sourced from self-provided reservoirs pumped from the digging wells. For the Baise outbreak location, a centralized water supply sourced from a mountain spring water well was used, and for the Huainan outbreak location, a centralized water supply sourced from a waterworks facility was used. We have rephrased the paragraph in **lines 135-140**.

Antimicrobial susceptibility tests showed that the outbreak isolates, including 149 isolates from patients and six from water samples, all had the cef^Razi^R phenotype, with additional resistance to ampicillin, ticarcillin, piperacillin, trimethoprim/sulfamethoxazole, gentamicin and tetracycline (Supplementary Table

S2). The *cef^Razi^R* isolates recovered from water samples and outbreak patients were similar in the Yulin, Baise, Nanning, and Guigang outbreaks, with only one to three different SNPs, suggesting that the contaminated water supply may have been the cause of the outbreaks (Figure 2).

We propose the following points as the reasons why most of the outbreak patients in this study were children:

- 1) The six outbreaks occurred mainly in kindergartens and schools, which were places where children gathered together.
- 2) These outbreaks occurred in southern China, where heavy precipitation and floods were frequent. During heavy precipitation and floods, the water supply systems, sewage systems, and waste disposal systems of the kindergartens and schools were damaged, resulting in contamination of drinking water sources. Meanwhile, some students in the kindergartens and schools would drink unboiled and unsterilized water directly, leading to an increased risk of infection among them.
- 3) According to previous studies, children under five years old are more susceptible to the infection of *Shigella* species.¹
- 4) It was found that most of the teaching staff did not take meals in the kindergartens and schools. Therefore, their risk of infection would be lower than that of students.

3. *Does these outbreaks repeatedly appeared in the same geographical location (more specifically the same school) during 2015-2020?*

Response: These outbreaks were independent, and appeared in different geographical locations. Five of the outbreaks occurred in Guangxi province, but in different schools in Yulin, Baise, Wuzhou, Nanning, and Guigang cities (Supplementary Table S1), and the remaining outbreak in Huainan was in Anhui province. Locations of the six outbreaks were marked on the map of the two provinces (Figure 1A).

4. *Introduction: explain the basics of the lineages (Hot et al., 2012) instead of abruptly citing the Lineage III).*

Response: Thank you for this comment. We have added the introduction of those lineages on the basis of the study of Holt et al:

“In 2012, Holt et al performed phylogenetic analysis on 132 globally distributed *S. sonnei* isolates, and the resulting phylogeny was divided into four distinct lineages (Lineage I to Lineage IV)².” (Lines 82-84)

5. *Line 112. Mention the period and source of isolation of these 152 sporadic strains from historical surveillance of shigellosis.*

Response: Thanks for your suggestion. We present the collection time of the sporadic strains in Figure 1B:

We have also described the collection time and isolation source of the sporadic strains in the revised manuscript:

“The sporadic strains were isolated from patients infected with *S. sonnei*, mainly during 2004-2016 (Figure 2 and Figure 1B).” (Lines 147-148)

6. *What is the rationale for using PFGE in this study? The plasmids would have been extracted directly from the isolates and tested for the AMR genes. The motioned 60-100 kbp fragment might contain chromosomal DNA fragment also. Of course, the Inc nature of the plasmids gives indirect evidence.*

Response: Thank you for this comment. In this study, S1-nuclease pulsed-field gel electrophoresis (PFGE) was performed to investigate the plasmid profiles of the *cef^Razi^R* and *col^R* strains, and Southern blotting was then used to determine the location of *bla_{CTX-M}*, *mphA*, and *mcr-1* genes on plasmids or chromosomes. The detailed method for S1-PFGE and Southern blotting was described by Zou D, et al³. The primer and probe sequences used in this study are listed in Supplementary Table S7. The S1-PFGE and Southern blotting results showed that the *bla_{CTX-M-14}* and *mphA* genes were co-located on the ~100 kbp plasmid, and the *mcr-1* gene was located on the ~60 kbp plasmid (Supplementary Figure S1).

We have rephrased the description of the methods for obtaining plasmids:

“To determine the *bla_{CTX-M-14}*-, *mphA*-, and *mcr-1*-harboring plasmid sequences, we extracted fragments of plasmids using a Qiagen Plasmid Midi kit (Qiagen, Germany) and submitted them for sequencing using the Illumina MiSeq platform to ensure that most of the fragments sequenced belonged to plasmids. We performed quality control using Trimmomatic and assembly using plasmidSPAdes⁴ on the plasmid sequence data before mapping them to the Plasmidfinder database⁵.” **(Lines 564-569)**

7. *Lines 170-172. Restructure the sentence. The AMR genes are more than the corresponding antimicrobials with the usage of the word “respectively”.*

Response: Thank you for the comment. According to your suggestion, we have revised the sentence:

“The isolates in Lineage III carried an average of 9.9 AMR genes, containing mostly *aac(3)-IIa*, *aadA1*, *aadA5*, *sat-2*, *strA*, and *strB* conferring resistance to aminoglycosides, *tet(A)* conferring resistance to tetracycline, *dfrA1* conferring resistance to trimethoprim, and *sul1* and *sul2* conferring resistance to sulfonamide.”
(Lines 214-217)

8. *Line 200-201. The 5.5 kb region with sul1, dfrA17, and aadA5 might be associated with an integron.*

Response: Thank you for your comment. The 5.5kb region was composed of the *intI1* integron and the *sul1*, *dfrA17*, and *aadA5* genes (Figure 4B).

In the revised manuscript, we have added the corresponding descriptions:

“the plasmids from the Baise, Nanning, Guigang, and Wuzhou outbreak isolates lost an IS26 copy and acquired a 5.5 kb region composed of the *intI1* integron and the *sul1*, *dfrA17*, and *aadA5* genes (Figure 4B).” **(Lines 253-255)**

“The WGS data and representative plasmids were mapped to the local INTEGRALL⁶ database to identify the type of integron using BLAST.” **(Lines 584-585)**

“We found that there was an *intI1*-type integron located downstream of the *dfrA17* gene associated with the cluster of *dfrA17*, *aadA5*, *qacEdelta*, and *sul1* (Supplementary Figure S4). In addition, there were four other gene clusters associated with integrons. The genes *tet(A)*, *strA*, *strB*, and *sul2* were mainly present in the order *sul2*, *tet(A)*, *strB*, and *strA*, with a 135 bp *intI1* fragment between the *tet(A)* and *strB* genes. Another combination of the *sul1*, *qacEdelta*, and *aadA1* genes had an *intI1* downstream of *aadA1*. The genes *dfrA1*, *sat-2*, and *aadA1* were arranged in the order *aadA1*, *sat-2*, and *dfrA1* with an *intI2* integron downstream of the *dfrA1* gene. There were still some strains carrying the *dfrA1* and *sat-2* genes with *intI2* downstream of *dfrA1* (Supplementary Figure S4).” **(Lines 221-229)**

9. Most of the AMR genes mentioned in lines 170-172, 205 might have located in the integrons. It is worth to identify the type of integrons, mostly the *IntI* and *IntI2* types. This information may also be included in the phylogenetic analysis.

Response: Thank you for this comment. There were 12 genes mentioned in lines 170-172 and 205 of the original manuscript. We found that there were mainly five combinations of 10 related genes.

We present this finding in Supplementary Figure S4:

Figure S4. Distribution, related resistance genes, and structures of integrons. A shows 10 resistance genes associated with integrons. B to F show the structures of the five integrons. ce^Razi^R coreistance to ceftriaxone and azithromycin.

We have also provided information on the types of integrons in the revised manuscript:

“We found that there was an *intI1*-type integron located downstream of the *dfrA17* gene associated with the cluster of *dfrA17*, *aadA5*, *qacEdelta*, and *sul1* (Supplementary Figure S4). In addition, there were four other gene clusters associated with integrons. The genes *tet(A)*, *strA*, *strB*, and *sul2* were mainly present in the order *sul2*, *tet(A)*, *strB*, and *strA*, with a 135 bp *intI1* fragment between the *tet(A)* and *strB* genes. Another combination of the *sul1*, *qacEdelta*, and *aadA1* genes had an *intI1* downstream of *aadA1*. The genes *dfrA1*, *sat-2*, and *aadA1* were arranged in the order *aadA1*, *sat-2*, and *dfrA1* with an *intI2* integron downstream of the *dfrA1* gene. There were still some strains carrying the *dfrA1* and *sat-2* genes with *intI2* downstream of *dfrA1* (Supplementary Figure S4).” (Lines 221-229)

10. Line 413. Mention stool DNA-based real-time polymerase chain reaction (RT-PCR) detection

Response: Thank you for your comment. We have revised the term accordingly in lines 489-490.

11. Most importantly, the virulence encoding genes are missing in the analysis. Epidemiologically, it is important to undertake this kind of analysis, as it is directly associated with the diseases. Authors can generate this formation using the WGS data.

Response: Thanks for your helpful comment. We have analyzed the virulence encoding genes and present the findings in Supplementary Figure S5:

Figure S5. Distribution of virulence-encoding genes with different functions. The exoenzyme, effector delivery system, motility, regulation, and two other function genes were absent in most of the strains. There were no significant associations between the genes and the cef^{RaziR} strains. cef^{RaziR} coresistance to ceftriaxone and azithromycin.

In the revised manuscript, we have also added and corrected the following descriptions:

“SNPs and genes associated with cef^{RaziR} isolates and outbreak strains” (**Line 270**)

“Analysis of potential markers and related genes” (**Line 587**)

“Considering that virulence genes may play certain roles, we also mapped WGS data to the VFDB⁷ using BLAST to identify the virulence-encoding genes and have displayed the results in a heatmap.” (Lines 601-603)

“Analysis of virulence-encoding genes showed that there were a total of 97 virulence factor genes, including 70 genes involved in the effector delivery system, three genes involved in exoenzyme activity, one involved in regulation, one involved in enterotoxin activity, one involved in motility, nineteen involved in nutritional or metabolic factor activity, and two involved in other functions. The genes of the effector delivery system, especially the T3SS, motility, regulation, and two other functions, were significantly different in distribution among the lineages. These virulence genes were absent in most of the *cef^Razi^R* strains but present in the strains from the Guigang outbreak and some of the Nanning outbreak strains (Supplementary Figure S5).” (Lines 286-294)

12. The SNPs analysis has to be extended in virulence encoding genes and its transcriptions and functions.

Response: Thank you for this helpful comment. We have extracted and analyzed SNPs located in the virulence factors genes of *S. sonnei* and supplemented Supplementary Table S6 to present the findings of this analysis.

We have added the findings of this analysis in the Results section:

“To further study the SNPs associated with virulence factors, we extracted all 173 SNPs located in virulence-encoding genes and found that these SNPs were located in 31 virulence genes. Among the SNPs, 91 were missense variations, and 82 were synonymous variations. The 91 missense variations affected 18 genes, including 11 nutritional or metabolic genes (*entA*, *entC*, *entD*, *entF*, *fepA*, *fepE*, *fepG*, *iucA*, *iutA*, *sitA*, and *sitD*), five effector delivery system genes (*ipaH_2*, *ipaH_3*, *rhsA*, *rhsB*, and *vgrG*), one exoenzyme gene (*sigA*), and one gene (*msbB*) with other functions (Supplementary Table S6).” (Lines 294-301)

13. *It is also suggested that the WGS data can be explored for MLST/MLVA/VNTR analysis to identify the sequence types.*

Response: Thanks for your suggestion. We have performed Multilocus sequence typing (MLST) analysis on the main Chinese clade strains and added the descriptions in the revised manuscript:

“MLST analysis was also conducted using MLST⁸ tools according to the database of the PubMLST website⁹.” (Lines 559-560)

“Multilocus sequence typing (MLST) analysis showed that all of the strains belonged to ST152, as previously described¹⁰.” (Lines 174-175)

14. *Discussion section mostly reflects the information provided in the results and this can be avoided.*

Response: Thank you for your comment. We have removed duplication with the results of the pangenome analysis and the GWAS (lines 335-345 in the original manuscript) from the Discussion section in the revised manuscript.

15. *Methodology for the isolation of shigellae from water/food samples was not mentioned.*

Response: Thanks for your comment. We have supplemented the detailed methods for the isolation of *Shigellae* from water/food samples and improved the description of methods:

“All food and water samples were collected by sterile sampling according to the national food safety standards¹¹, such as GB 4789.1-2016, GB 4789.2-2016, GB 4789.3-2016, GB 4789.4-2016, GB 4789.6-2016, and GB 4789.10-2016, and the standards for drinking water quality (GB 5749-2006)¹². The samples were screened for *Shigella* spp. using RT-PCR assays (Shanghai BioGerm Medical Biotechnology

Co., Ltd., Shanghai, China), and suspected pathogenic bacteria were isolated through standard procedures by local CDCs. Water samples were filtered through a 0.45 µm sterile membrane (Millipore, USA) using a vacuum filtration apparatus and then inoculated on xylose lysine deoxycholate (XLD) agar plates, *Salmonella-Shigella* agar (SS agar), and/or *Salmonella* chromogenic agar plates (Beijing Land Bridge Technology Co., Ltd., China). Suspected isolates recovered from the samples were identified using a commercial biochemical test kit (API 20E system; bioMérieux Vitek, France) according to the manufacturer's recommendations.” (Lines 493-505)

16. *Reference numbers mentioned in the text and in the concerned section are not matching.*

Response: We are very sorry for the negligence. We have updated reference numbers throughout the text to match those in the References section.

17. *Finally, the English language has to be improved. In many places, grammar and syntax errors exists.*

Response: The manuscript has been thoroughly revised and polished by native speakers, and we hope it can meet the journal's standards. Thank you so much for your useful comments.

Responses to Reviewer #2

1. *This study reports on the sampling and genomic epidemiology investigation of a large sample of outbreak- and sporadic isolates of S. sonnei collected in China. Applied methods are scientifically sound. The Chinese isolates are analyzed in context of a global S. sonnei strain panel and the previously established lineages to assess the genome plasticity and differentiating features. The in-depth analysis discusses chromosome- and plasmid content and inventories along with the respective XDR profiles. Overlaying meteorological data is a strength.*

Response: Thank you for your positive comments.

- 2. The discussion is to some extent redundant and reiterates the results e.g. SNP, COG, and plasmids compositions in granular detail; it should be more concise to highlight and interpret key findings.*

Response: Thank you for your suggestion. We have removed the duplication with the results of the pangenome analysis and the GWAS (lines 335-345 in the original manuscript) from the Discussion section and have added the interpretation of the COG analysis in the revised manuscript.

- 3. The two COG categorizations are somewhat superficial compared the level of detail presented elsewhere and is not particularly insightful. These data should maybe be discussed in context to what is observed in the other lineages/other outbreaks.*

Response: Thank you for your suggestion. We have interpreted and discussed the results of the COG analysis in detail in the revised manuscript:

“The COG results of phenotype-related pangenomic genes and SNP sites showed that all these genes involved were related to energy production; carbohydrate transport and metabolism; transcription, replication, recombination, and repair; cell wall or membrane or envelope biogenesis, and other functions. Although the effects of these related genes on the biological functions of the strains cannot be determined by simple mathematical addition of the number of genes involved in these two COGs, there were many more genes associated with transport and energy production, such as the *fep* gene encoding iron enterobactin, *msb* gene encoding a full transporter, and *tauB* gene encoding the taurine ATP-binding component of a transport system^{13,14}. We hypothesize that the strains with the *cef^Razi^R* phenotype may have stronger nutrient transport and metabolic capacity, and to some extent, this may contribute to better

adaptation to the environment of these strains. In addition, cell wall and membrane and envelope biogenesis¹⁵-associated genes, such as the phage-related lysozyme gene *rrrD*, may play a role in adaptation to poor environments. These may be important reasons why certain waterborne strains can survive in water and contribute to a series of outbreaks.” (Lines 408-421)

4. *Use of the "first time/study" seems a bit overstated throughout the MS given the novelty of original research.*

Response: Thank you for this comment. We have deleted the inappropriate “first time/study” from the revised manuscript.

5. *Some methods need clarification.*

Response: Thanks for your comment. We have supplemented detailed information about the methods of isolating strains, the tools used in correcting errors, and the analyses on genotyping, multi-locus sequence typing, plasmids comparisons, integrons, and virulence encoding genes:

“All food and water samples were collected by sterile sampling according to the national food safety standards¹¹, such as GB 4789.1-2016, GB 4789.2-2016, GB 4789.3-2016, GB 4789.4-2016, GB 4789.6-2016, and GB 4789.10-2016, and the standards for drinking water quality (GB 5749-2006)¹². The samples were screened for *Shigella* spp. using RT-PCR assays (Shanghai BioGerm Medical Biotechnology Co., Ltd., Shanghai, China), and suspected pathogenic bacteria were isolated through standard procedures by local CDCs. Water samples were filtered through a 0.45 µm sterile membrane (Millipore, USA) using a vacuum filtration apparatus and then inoculated on xylose lysine deoxycholate (XLD) agar plates, *Salmonella-Shigella* agar (SS agar), and/or *Salmonella* chromogenic agar plates (Beijing Land Bridge Technology Co., Ltd., China). Suspected isolates recovered from the samples were

identified using a commercial biochemical test kit (API 20E system; bioMérieux Vitek, France) according to the manufacturer's recommendations.” (Lines 493-505)

“De novo assemblies were generated using Canu v1.6,¹⁶ and errors of sequencing were corrected by the Illumina sequencing data using Pilon v1.24¹⁷.” (Lines 537-538)

“Further genotyping of the strains was performed using Mykrobe software according to the study of Jane Hawkey et al.¹⁰” (Lines 558-559)

“MLST analysis was also conducted using MLST⁸ tools according to the database of the PubMLST website⁹.” (Lines 559-560)

“Plasmid sequence comparison and map generation were performed using BLAST¹⁸ and custom Perl scripts, respectively.” (Lines 573-575)

“The WGS data and representative plasmids were mapped to the local INTEGRALL⁶ database to identify the type of integron using BLAST.” (Lines 584-585)

“Considering that virulence genes may play certain roles, we also mapped WGS data to the VFDB⁷ using BLAST to identify the virulence-encoding genes and have displayed the results in a heatmap.” (Lines 601-603)

6. *Overall this work will contribute to the global understanding of shigella pathogenome evolution and the AR virulence traits of emerging clones that pose public health threats.*

Response: Thank you for your positive comments.

7. *Line 41: report (I assume the statement refers to the current study with the additional AR profile elucidated and not the previous work - citation 13)*

Response: We are deeply sorry for the errors here and below. We appreciate your patience in helping us improve our writing. We have made the suggested corrections here (line 39) and below (lines 318, 397, and 451).

8. *Line 91: a plasmid*

Response: We have corrected the error in **line 99**:

“some of the outbreak isolates had even acquired a plasmid-mediated *mcr-1* gene”

9. *Line 115: the cefR*

Response: Thank you for this comment. We have corrected it in **line 151**:

“Antimicrobial susceptibility tests showed that the outbreak isolates, including 149 isolates from patients and six from water samples, all had the $\text{cef}^{\text{R}}\text{azi}^{\text{R}}$ phenotype”

10. *Line 135: The majority of Chinese*

Response: Thank you for this comment. We have revised it in **lines 171-172**:

“The majority of the Chinese isolates were located in Lineage III and formed a distinct Chinese clade (main Chinese clade).”

11. *Line 138: demonstrating > suggesting*

Response: Thank you for this comment. We have revised it in **line 177**:

“It is estimated that the main Chinese *S. sonnei* clade shares a most recent common ancestor (MRCA) that existed circa 1987 (Figure 1C), suggesting a recently derived clone of *S. sonnei* circulating in China.”

12. *Line 141: may have emerged*

Response: Thank you for this comment. We have revised it in **line 180**:

“The outbreak isolates formed a recently derived monophyletic group that may have emerged circa 2010 (Figure 1C and Figure 2).”

13. Line 155: Were these relationships expected based on geography/flooding etc.
(Fig 1)

Response: Thank you for this question. To answer the question, we have specifically explored the microevolution of cef^Razi^R strains and added the corresponding description in the revised manuscript:

“The strains in the Huainan outbreak were closer to those in the Yulin outbreak, with 46 SNPs in the 49 strains. The strains from Baise were closer to those from Guigang, with 22 SNPs in 44 strains, and the strains from Nanning were closer to those from Wuzhou, with 32 SNPs among the 62 strains. The geographical locations of the five outbreaks in Guangxi from west to the east are in the order Baise, Nanning, Guigang, Yulin, and Wuzhou (Figure 1A and Supplementary Figure S2). These five cities are located in the Pearl River Basin, and the Pearl River flows from southwest to northeast. The location of Huainan is near the Yangtze River Basin and Huaihe Basin.” (Lines 194-201)

14. Line 179: Why is the acquisition of yet another resistance surprising?

Response: Our laboratory previously reported several *S. sonnei* strains carrying a plasmid-mediated *mcr-1* gene.¹⁹ The *mcr-1* gene was horizontally transferred via IncI2 plasmids. These IncI2 plasmids were similar to the pHNSHP45 (accession no.: KP347127) from the *E. coli* strain SHP45 from a pig farm in Shanghai²⁰ and the pEG430-1 (accession no.: LT174530) from an *mcr-1*-positive *S. sonnei* strain in Vietnam²¹. Before the Nanning outbreak in the present study, no outbreaks caused by MDR *S. sonnei* harboring the *mcr-1* gene conferring resistance to colistin have been reported. This may indicate that the colistin-resistant strains have spread among the population and even caused epidemics in some areas. The strains with more drug-resistant agents would bring greater difficulties to clinical treatment and even affect the development of prevention and control strategies for *Shigella*. Therefore,

we think that the acquisition of colistin resistance by *S. sonnei* strains is “surprising”.

15. *Line 185: phenotypes*

Response: Thank you for this comment. We have corrected it in **line 238**:

“Plasmid profiles and comparisons of plasmids associated with the $\text{cef}^{\text{R}}\text{azi}^{\text{R}}$ and col^{R} phenotypes”

16. *Line 189: rephrase*

Response: We appreciate your valuable comment. We have rephrased the sentence to make it easier to understand:

“With the Illumina and Nanopore sequencing platforms, 31 complete plasmids were obtained in this study, and these plasmids harbored one or more of the following resistance genes: *bla*_{CTX-M}, *bla*_{CMY-2}, *bla*_{TEM}, *mphA/ermB*, and *mcr-1* (Supplementary Table S4).” (**Lines 242-244**)

17. *Line 203: the backbone*

Response: Thank you for this comment. We have corrected the grammar error here and the same errors elsewhere. (**Lines 257-260**)

18. *Line 205: the backbone*

Response: Thank you for this comment. We have corrected the grammar error here and the same errors elsewhere. (**Lines 257-260**)

19. *Line 222: elaborate on "almost the same"*

Response: There were a total of 215 cef^Razi^R isolates in our study, including 155 outbreak isolates. The outbreaks isolates were an important component of these cef^Razi^R isolates. We performed a GWAS on the outbreak and the cef^Razi^R isolates. A total of 59 significant SNPs were associated with the outbreak isolates (Figure 5A) and 60 significant SNPs were associated with the cef^Razi^R isolates (Figure 5B). Among these SNPs, only three were different, including one same-sense mutation, one in *gcd*, and one in *fhuE* (Supplementary Table S5). Therefore, we concluded that associated SNPs with the outbreak and the cef^Razi^R isolates were almost the same. To some degree, this revealed that the SNPs found in outbreak isolates also had a strong correlation with other cef^Razi^R isolates. These SNPs may play an important role in transportation and energy production.¹³

We have elaborated on "almost the same" in the revised manuscript:

“In addition to the 10 highly significant SNPs, other SNPs found in the cef^Razi^R isolates were almost the same as those in the outbreak isolates, with only three SNPs different (one same-sense mutation, one in *gcd*, and one in *fhuE*) (Supplementary Table S5).” (Lines 276-279)

20. Line 243: have been rarely

Response: Thank you for this comment. We have revised it in **line 316**:

“Nevertheless, outbreaks caused by MDR *S. sonnei* with cef^Razi^R resistance have rarely been reported.^{22,23}”

21. Line 245: report

Response: Thank you for this comment. We have revised it in **line 318**:

“Concerningly, here, we report the prevalence and outbreaks of shigellosis caused by *S. sonnei* with a combination of MDR, cef^Razi^R, reduced susceptibility to fluoroquinolones, and even col^R.”

22. *Line 257: may have emerged*

Response: Thank you for this comment. We have revised it in **line 330**:

“Our data showed that the Chinese *S. sonnei* isolates formed a main Chinese clade within the global Lineage III that may have emerged circa 1987.”

23. *Line 307: locally, in the local population*

Response: Thank you for this comment. We have revised it in **lines 380-381**:

“These results indicated that this distinct clone harboring the *bla*_{CTX-M-14} and *mphA*-positive plasmids might have been prevalent in the local populations and spread interregionally.”

24. *Line 322: rephrase ever reported*

Response: We appreciate the valuable comment. We have rewritten the sentence to make it easier to understand:

“We have reported the emergence of *mcr-I*-positive *S. sonnei* strains in China,¹⁹ and here, we report the first outbreak of shigellosis caused by *mcr-I*-positive *S. sonnei* strains with the *cef*^R*azi*^R phenotype.” (**Lines 396-398**)

25. *Line 328: in the process > towards?*

Response: Thank you for this comment. We have revised it in **line 402**:

“these distinct plasmids might have been prevalent in the local populations and spread interregionally and seem to be evolving toward local establishment.”

26. *Line 333: Suggest to delete> ... than just the treatment of Shigellosis.*

Response: Thank you for the helpful suggestion. We have deleted these redundant words and revised the sentence from “which would represent a major threat to global public health and thus have a much broader implication than just the treatment of shigellosis” to “which would represent a major threat to global public health and thus have much broader implications.” (Lines 405-406)

27. *Line 336: While... ? check intended meaning of this sentence.*

Response: Thank you for pointing out this problem. This sentence has been removed from the revised manuscript since the paragraph in which the sentence is located overlapped with the Results section.

28. *Line 380: globalization*

Response: Thank you for this comment. We have corrected it in **line 456**:

“Coupled with globalization and the increased movement of people across national borders, this particular clone has the potential to spread globally, posing a risk of the spread of highly resistant shigellosis.”

29. *Line 404: prompted*

Response: Thank you for this comment. We have corrected it in **line 480**:

“This alarming trend prompted us to investigate the six outbreaks and explore the link between the outbreak and the historical strains recovered from our routine surveillance of shigellosis.”

30. *Line 405: outbreak and historical strains*

Response: Thank you for this comment. We have corrected it in **line 481**:

“This alarming trend prompted us to investigate the six outbreaks and explore the link between the outbreak and the historical strains recovered from our routine surveillance of shigellosis.”

31. *Line 417: are presented > were considered?*

Response: Thank you for this comment. We have corrected it in **line 493**:

“The meteorological data, especially the annual precipitation and flood disaster data collected from the National Bureau of Statistics of China (<http://www.stats.gov.cn/>), were also considered.”

32. *Line 452: errors of sequence were corrected ... Which tool was used for error correction?*

Response: Thank you for this comment. We have added the tool used in assembly and error correction:

“De novo assemblies were generated using Canu v1.6,¹⁶ and errors of sequencing were corrected by the Illumina sequencing data using Pilon v1.24¹⁷.” (**Lines 537-538**)

33. *Line 476: Plasmid sequence data were performed for quality control ... Please rephrase.*

Response: Thank you for your kind suggestion. We have rephrased the sentence to avoid confusion:

“We performed quality control using Trimmomatic and assembly using plasmidSPAdes⁴ on the plasmid sequence data before mapping them to the Plasmidfinder database⁵.” (**Lines 568-569**)

34. *Line 482: Plasmid sequence comparison and map generation were performed using BLAST, respectively. Which tool was used to generate the linear plasmid displays?*

Response: Thank you for this comment. We apologized for our negligence in missing the introduction of the tool used. We have revised the description:

“Plasmid sequence comparison and map generation were performed using BLAST¹⁸ and custom Perl scripts, respectively.” (**Lines 573-575**)

35. *Line 507: analyses*

Response: Thank you for your helpful comments. We have corrected it in **line 600**:

“COGs of genes identified by pangenome analysis and a GWAS were determined by mapping to the COG function database using BLAST and visualized using R.”

Responses to Reviewer #3

1. *The manuscript titled "An extensively drug-resistant clone of Shigella sonnei with resistance to azithromycin, ceftriaxone, and/or colistin associated with waterborne outbreaks of shigellosis in China" used whole-genome sequencing to describe XDR S. sonnei from outbreaks of diarrhea in China. The authors compared isolates from six shigellosis outbreaks in China to sporadic isolates from China as well as previously described global S. sonnei, providing important insight into the expansion of this XDR clone in China.*

Response: Thank you for the positive comments.

2. *XDR and MDR: How were the isolates determined to be XDR and MDR? How many antimicrobial categories did the isolates exhibit resistance to? Please define in the introduction and/or methods.*

Response: Thank you for this comment. We have added the definitions of XDR and MDR strains in the Methods section:

“In this study, MDR strains were defined as those with resistance to three or more categories of antibiotics, and XDR was defined as MDR with an additional resistance to azithromycin, ceftriaxone, or colistin.” **(Lines 517-519)**

3. *Line 41: Change "reported" to "report". Also, it is not clear what is meant by a distinct phylogenetic group. How does the XDR clone fit within the genotyping scheme described by Hawkey et al. 2021 (PMID: 33976138)?*

Response: Thank you for this comment. We appreciate your patience in helping us improve our writing. We have revised the grammar errors here **(line 39)** and below **(lines 318, 397, and 451)**.

We used the software Mykrobe¹⁰ to identify genotyping of the strains in our study, and we found that most of the Chinese strains belonged to the 3.7.6 genotype, corresponding to global Lineage III. The findings were presented in Figure 1C. We have added the corresponding descriptions in the Methods and Results sections:

“Further genotyping of the strains was performed using Mykrobe software according to the study of Jane Hawkey et al.¹⁰” **(Lines 558-559)**

“Further genotyping of the main Chinese clade strains showed that they belonged to the 3.7.6 genotype, corresponding to global Lineage III (Figure 1C).” **(Lines 173-174)**

4. *Line 45: What is the number of outbreaks?*

Response: Thank you for this comment. There were six outbreaks in our study during 2015-2020. We have changed “a series of” to “six” in **line 43**.

5. *Line 48: Change "with mcr-1 gene" to "with an mcr-1 gene".*

Response: Thank you for pointing out this error. We have added "an" before the "*mcr-1 gene*" in **line 49**:

“The cef^Razi^R and col^R phenotypes were attributed to the acquisition of diverse distinct plasmids, particularly the IncB/O/K/Z plasmid coharboring the *bla*_{CTX-M-14}, *mphA*, *aac(3)-IId*, *dfrA17*, *aadA5*, and *sulI* genes and the IncI2 plasmid with an *mcr-1* gene.”

6. *Line 72: Please add a reference.*

Response: According to your comment, we have added a reference to support our statement:

“Antimicrobials, especially fluoroquinolones (e.g., ciprofloxacin and norfloxacin), third-generation cephalosporins (e.g., ceftriaxone), and macrolides (e.g., azithromycin), are recommended by the World Health Organization (WHO) for the treatment of shigellosis to accelerate recovery, prevent complications, and reduce onward transmission.²⁴” (**Lines 69-72**)

7. *Line 74: Change "locates" to "located" and "Global III Lineage" to "global Lineage III".*

Response: Thank you for your valuable comment. We have accordingly revised the sentence in **line 74**:

“this pandemic clone is located within global Lineage III, which recently emerged from Europe and underwent contemporary global dispersal and localized clonal expansion.^{2,25}”

8. *Lines 77-78: Change to "has been increasingly detected among MDR S. sonnei"*

Response: Thank you for the suggestion. The sentence has been revised in **line 78**:

“Moreover, resistance to fluoroquinolones has been increasingly detected among MDR *S. sonnei*”

9. *Line 82: Change "within" to "at".*

Response: We are sorry for this mistake. We have reworded the sentence according to your comment in **line 84**:

“Several studies have investigated the genomic epidemiology of *S. sonnei* at the global and regional levels”

10. *Lines 84-85: Please elaborate on why these groups have great public health significance.*

Response: Thank you for this comment. We have elaborated on the public health significance of the phylogenetic groups of Lineage III in the revised manuscript:

“These studies on *S. sonnei* strains from the UK and France²⁶, Latin America and the Caribbean²⁷, Vietnam²⁵, Bhutan, Australia, and Ireland²⁸ showed that most of these newly disseminated *S. sonnei* groups belonged to a single, globally distributed, multidrug-resistant clade of *S. sonnei* Lineage III, which was referred to as global Lineage III. Furthermore, these groups are associated with features such as ciprofloxacin resistance, azithromycin resistance, and transmission among men who have sex with men (MSM).^{10,26,28} These results reveal that Lineage III has become a popular branch of MDR *S. sonnei*, posing a great threat to public health in these countries and even around the world.” **(Lines 87-94)**

11. *Lines 90-92: Are these recent outbreaks first described in this study? Or can references be provided where these have been described in other publications?*

Response: Thank you for this comment. An outbreak in Yulin city of Guangxi province in 2015 was previously reported by us, which was also included in the six outbreaks of the present study. We have clarified this in the revised manuscript and have cited the 2015 outbreak:

“We reported a waterborne shigellosis outbreak caused by MDR *S. sonnei* with resistance to ceftriaxone and azithromycin (cef^Razi^R) that occurred in 2015.²² Since then, we have identified the prevalence of MDR *S. sonnei* with cef^Razi^R and reduced susceptibility to fluoroquinolones, which has caused five additional waterborne shigellosis outbreaks in China.” (Lines 94-98)

12. *Line 93: Should this be "with concurrent resistance to"?*

Response: Thank you for your constructive comment. Following your comment, we have made the revision in **lines 100-101**:

“The emergence and prevalence of MDR *S. sonnei* isolates with concurrent resistance to ciprofloxacin, azithromycin, ceftriaxone, and/or colistin will inevitably greatly narrow the choice of effective antimicrobials, especially among children”

13. *Lines 95-96: Please provide references.*

Response: Thank you for your comment. We have provided a reference for the statement:

“considering the cartilage toxicity of fluoroquinolones, ceftriaxone and azithromycin are recommended as alternative treatments for children²⁹” (Lines 102-104)

14. *Line 104: Additional details of the outbreaks such as age of the patients and years would be informative to include here. I would suggest moving the first two sentences from the supplementary results to this section.*

Response: Thank you for this helpful suggestion. We have moved the description of patients in outbreaks from supplementary results to the Results section of the revised manuscript:

“The majority (92.4%) of these patients were under 14 years old, and male patients accounted for 48.3%. They all showed symptoms of diarrhea, followed by fever (94.0%), abdominal pain (70.2%), vomiting (67.7%), rectal tenesmus (50.3%), nausea (22.2%), and dizziness (1.6%) (Supplementary Table S1). All these patients recovered, with no deaths.” **(Lines 116-120)**

15. *Line 141: Should this say "may have emerged"?*

Response: Thank you. We have corrected this error in **line 180**:

“The outbreak isolates formed a recently derived monophyletic group that may have emerged circa 2010 (Figure 1C and Figure 2).”

16. *Line 146: Is this referencing the supplementary data table? Or the analyses described in the supplementary results? Please clarify.*

Response: Thank you. Here we intend to refer to the supplementary data table, which has been renamed Supplementary Table S3 in the revised manuscript. We have revised this sentence for clarification:

“As shown in Supplementary Table S3, our dataset contains genome sequences with different spatiotemporal and phenotypic contexts” **(Lines 184-185)**

17. *Line 169: Should these be Figures S1A and S3?*

Response: Thank you. We apologize for the confusion. In lines 168-169 of the original manuscript, we did refer to Extended Data Figure 1A and Figure 3. Nonetheless, as per your comment below, we have included Extended Data Figure 1 in supplementary figures, which is now Supplementary Figure S3. Therefore, we have

updated the referenced figures to Supplementary Figure S3A and Figure 3 in **lines 213-214** of the revised manuscript.

18. *Lines 186-187: Should this be "four plasmids representing different replicons"?*

Response: Thank you for this comment. We apologize for the ambiguity in our statement and we have corrected “four replicons of plasmids” to “four plasmids representing different replicons” in the sentence. (**Lines 239-240**)

19. *Line 202: Should this be "three distinct plasmid replicons"?*

Response: Thank you for this comment. We apologize for the ambiguity in our statement and we have corrected “three replicons of plasmid backbones” to “three distinct plasmid replicons” (**Line 256**)

20. *Line 217: For consistency the $-\log_{10}(P)$ should be represented as p-value as written later in this section.*

Response: Thank you for this comment. We have corrected “ $-\log_{10}(P) > 100$ ” and “ $-\log_{10}(P) > 50$ ” to “ $P < 1e-100$ ” and “ $P < 1e-50$ ”, respectively, in the revised manuscript. (**Lines 272-273 and 275**)

21. *Lines 236-239: How are these mutations potentially significant to expansion of the MDR clones? This can be elaborated upon in the discussion (see comment, Lines 335-345).*

Response: Thank you for your comments. We have elaborated on the significance of these mutations to the expansion of the MDR clones in the Discussion section:

“The COG results of phenotype-related pangenomic genes and SNP sites showed that all these genes involved were related to energy production; carbohydrate

transport and metabolism; transcription, replication, recombination, and repair; cell wall or membrane or envelope biogenesis, and other functions. Although the effects of these related genes on the biological functions of the strains cannot be determined by simple mathematical addition of the number of genes involved in these two COGs, there were many more genes associated with transport and energy production, such as the *fep* gene encoding iron enterobactin, *msb* gene encoding a full transporter, and *tauB* gene encoding the taurine ATP-binding component of a transport system^{13,14}. We hypothesize that the strains with the *cef^Razi^R* phenotype may have stronger nutrient transport and metabolic capacity, and to some extent, this may contribute to better adaptation to the environment of these strains. In addition, cell wall and membrane and envelope biogenesis¹⁵-associated genes, such as the phage-related lysozyme gene *rrrD*, may play a role in adaptation to poor environments. These may be important reasons why certain waterborne strains can survive in water and contribute to a series of outbreaks.” (Lines 408-421)

22. Lines 247-251: Is this description also based on published findings? Please provide the references.

Response: Thank you for the comment. We have added supporting references in this sentence:

“*S. sonnei* clones belonging to Lineage III appear to be more easily transmitted from person to person and spread across regions²⁶ and to be highly proficient at acquiring resistance to additional antimicrobials, such as fluoroquinolones, macrolides, third-generation cephalosporins, and polymyxins, when they are introduced into new locations^{30,31}.” (Lines 320-323)

23. Line 257: Should this be "may have emerged"?

Response: Thank you for this comment. We have corrected it in **line 330**:

“Our data showed that the Chinese *S. sonnei* isolates formed a main Chinese clade within the global Lineage III that may have emerged circa 1987.”

24. *Line 271: Change "antibiotics" to "antibiotic".*

Response: Thank you. We have corrected it in **line 344**:

“The severe MDR of this specific clone may be due to the strong selective pressures imposed by antibiotic usage.”

25. *Line 307: Should this be "might have been prevalent locally"? Or in "local populations"?*

Response: Thank you. We have corrected “in the local” to “in the local populations” in **lines 380-381**.

26. *Line 322: It is not clear what is meant by "We had ever reported". Please clarify.*

Response: We appreciate your valuable comment. We have rewritten the sentence to make it easier to understand:

“We have reported the emergence of *mcr-1*-positive *S. sonnei* strains in China,¹⁹ and here, we report the first outbreak of shigellosis caused by *mcr-1*-positive *S. sonnei* strains with the cef^Razi^R phenotype.” (**Lines 396-398**)

27. *Lines 335-345: These lines are repetitious with what is in the results and much of this detail could be removed to help shorten the discussion. Provide more interpretation of how mutations in these types of genes may facilitate the expansion of the MDR isolates.*

Response: Thank you for your comment. We have removed the duplication with the results of the pangenome analysis and the GWAS (lines 335-345 in the original

manuscript) from the Discussion section and have elaborated on the significance of mutations to the expansion of the MDR clones in the Discussion section of the revised manuscript:

“The COG results of phenotype-related pangenomic genes and SNP sites showed that all these genes involved were related to energy production; carbohydrate transport and metabolism; transcription, replication, recombination, and repair; cell wall or membrane or envelope biogenesis, and other functions. Although the effects of these related genes on the biological functions of the strains cannot be determined by simple mathematical addition of the number of genes involved in these two COGs, there were many more genes associated with transport and energy production, such as the *fep* gene encoding iron enterobactin, *msb* gene encoding a full transporter, and *tauB* gene encoding the taurine ATP-binding component of a transport system^{13,14}. We hypothesize that the strains with the *cef^Razi^R* phenotype may have stronger nutrient transport and metabolic capacity, and to some extent, this may contribute to better adaptation to the environment of these strains. In addition, cell wall and membrane and envelope biogenesis¹⁵-associated genes, such as the phage-related lysozyme gene *rrrD*, may play a role in adaptation to poor environments. These may be important reasons why certain waterborne strains can survive in water and contribute to a series of outbreaks.” **(Lines 408-421)**

28. *Lines 374-375: Remove MDR as XDR are by definition MDR. It would be informative here to describe the number of antibiotic categories to which the XDR clone exhibits resistance.*

Response: Thank you. We have revised the description of XDR *S. sonnei* clones here according to your comment:

“In conclusion, we report an XDR *S. sonnei* clone with *cef^Razi^R*, reduced susceptibility to fluoroquinolones and even *col^R*” **(Lines 451-452)**

29. *Line 382: Does this mean the potential to spread globally?*

Response: Yes, here we intend to mean “the potential to spread globally”. This sentence has been revised:

“Coupled with globalization and the increased movement of people across national borders, this particular clone has the potential to spread globally, posing a risk of the spread of highly resistant shigellosis.” (Lines 456-458)

30. Lines 490-492: This sentence is not clear. Please revise.

Response: Thank you for this comment. We have rewritten the sentence to make it more understandable:

“In addition, if the assembly sequences of the strains included AMR genes that were harbored in corresponding plasmids, we concluded that the strains carried the same plasmids or plasmids with a similar structure.” (Lines 582-584)

31. Line 529: Were the genome assemblies and sequence reads of each isolate analyzed by WGS in this study deposited into a public repository such as GenBank? This should be described here and accession numbers of assemblies and SRAs should be added to the isolate table.

Response: Thank you for your suggestion. We have submitted the sequence reads of each isolate to NCBI's Sequence Read Archive (SRA) under BioProject number PRJNA835603. We have clearly stated this in the “Data Availability” section of the revised manuscript (line 615) and provided the accession numbers in Supplementary Table S3.

32. Fig S1: How were the ladder images generated that are overlayed on the southern blots? Were the ladders from an inverse exposure of panel A? In the figure legend please mention the ladder used and describe that panels B, C, and D contain the ladders from the ethidium-stained gel prior to transfer, overlayed on

the southern hybridization images. Also, please add the ladder band sizes to the bottom row of images.

Response: Thank you for this comment. Panel A shows the pulsed-field gel electrophoresis (PFGE) of the nuclease obtained from isolates. *Salmonella enterica* serotype Braenderup H9812 digested with XbaI was used as the size marker for the PFGE and southern. The size of the Nylon membrane was the same as the PFGE. We obtained two types of plasmids of ~100kbp and ~60kbp. We then extracted corresponding plasmids for the Southern hybridization of resistance genes. Panel B shows that the plasmids of ~60kbp carried the *mcr-1* gene. Panels C and D show that the plasmids of ~100kbp carried the *bla*_{CTX-M-14} and *mphA* genes.

Following your suggestion, we have added corresponding markers to panels B, C, and D of the revised Supplementary Figure S1 and revised its legend:

Figure S1. S1-PFGE and Southern blotting analysis of the *mcr-1*-, *bla*_{CTX-M-14}, and *mphA*-positive plasmids. Eleven representative isolates, including ten from patients and one from tap water, were selected for S1-PFGE and Southern hybridization analysis. The universal standard strain *Salmonella braenderup* H9812 digested with XbaI was used as the size marker for the PFGE and southern. The size of the Nylon membrane was the same as the PFGE. **A** S1 nuclease plasmid profile obtained by PFGE. **B** Southern hybridization for the *mcr-1* gene. **C** Southern hybridization for the *bla*_{CTX-M-14} gene. **D** Southern hybridization for the *mphA* gene. S1-PFGE S1 nuclease pulsed-field gel electrophoresis.

33. *Extended Data Figure 1 and Extended Data Table 1: It is confusing to have a supplemental Figure S1, and also an Extended Data Figure 1. Can the extended data table and figure be included among the supplementary tables and figures?*

Response: Thank you for your comments to help us improve the quality of this paper. In the revised manuscript, we have included Extended Data Figure 1 and Extended Data Table 1 in the Supplementary Information as Figure S3 and Table S2, respectively.

As suggested, we have added Supplementary Figure S4 to present the distribution, related resistance genes, and structures of integrons, Supplementary Figure S5 to present the distribution of virulence-encoding genes with different functions, and Supplementary Table S6 to present the SNPs associated with virulence encoding genes in the revised manuscript. The Supplementary Data presenting the basic information of the genome-sequenced strains used in this study have been included in the Supplementary Information as Table S3. Accordingly, we have updated the numbering of figures and tables throughout the revised manuscript:

Figures

Figure 1. Geographic distribution and phylogenetic analysis of Chinese *S. sonnei* strains, including the outbreak strains

Figure 2. Microevolution of the *cef*^R*azi*^R *S. sonnei* strains including the outbreak strains

Figure 3. Phylogeny of the global *S. sonnei* isolates with a heatmap showing the distribution of AMR determinants

Figure 4. Plasmid profiles and pairwise comparisons of plasmids associated with the $\text{cef}^{\text{R}}\text{azi}^{\text{R}}$ and col^{R} phenotypes

Figure 5. The SNPs identified by a GWAS are associated with $\text{cef}^{\text{R}}\text{azi}^{\text{R}}$ isolates and the outbreak strains

Figure 6. Accessory genes identified by pangenome analysis that might be associated with $\text{cef}^{\text{R}}\text{azi}^{\text{R}}$ isolates and the outbreak strains

Supplementary figures

Figure S1. S1–PFGE and Southern blotting analysis of the *mcr-1*-, *bla*_{CTX-M-14}, and *mphA*-positive plasmids

Figure S2. Annual precipitation and flood-affected areas in Guangxi and Anhui Provinces from 2011 to 2020

Figure S3. Box plot representing the antimicrobial resistance gene content among different groups

Figure S4. Distribution, related resistance genes, and structures of integrons

Figure S5. Distribution of virulence-encoding genes with different functions

Figure S6. Pangenome analysis of global *S. sonnei* isolates

Figure S7. Temporal distribution of cases from the outbreaks

Supplementary tables

Table S1. Characteristics of the patients from the six outbreaks

Table S2. Antimicrobial resistance of the Chinese *S. sonnei* isolates including the outbreak strains

Table S3. The genome-sequenced *S. sonnei* isolates used in this study

Table S4. The plasmids identified in this study

Table S5. The SNPs detected associated with $\text{cef}^{\text{R}}\text{azi}^{\text{R}}$ isolates especially the outbreak strains

Table S6. The SNPs associated with virulence encoding genes

Table S7. The accessory genes identified associated with cef^Razi^R isolates, especially the outbreak strains

Table S8. The primers used in this study

References

- 1 Zaidi, M. B. & Estrada-García, T. Shigella: A Highly Virulent and Elusive Pathogen. *Curr Trop Med Rep* **1**, 81-87, doi:10.1007/s40475-014-0019-6 (2014).
- 2 Holt, K. E. *et al.* Shigella sonnei genome sequencing and phylogenetic analysis indicate recent global dissemination from Europe. *Nat Genet* **44**, 1056-1059, doi:10.1038/ng.2369 (2012).
- 3 Zou, D. *et al.* A novel New Delhi metallo-β-lactamase variant, NDM-14, isolated in a Chinese Hospital possesses increased enzymatic activity against carbapenems. *Antimicrob Agents Chemother* **59**, 2450-2453, doi:10.1128/aac.05168-14 (2015).
- 4 Antipov, D. *et al.* plasmidSPAdes: assembling plasmids from whole genome sequencing data. *Bioinformatics* **32**, 3380-3387, doi:10.1093/bioinformatics/btw493 (2016).
- 5 Carattoli, A. & Hasman, H. PlasmidFinder and In Silico pMLST: Identification and Typing of Plasmid Replicons in Whole-Genome Sequencing (WGS). *Methods Mol Biol* **2075**, 285-294, doi:10.1007/978-1-4939-9877-7_20 (2020).
- 6 Moura, A. *et al.* INTEGRALL: a database and search engine for integrons, integrases and gene cassettes. *Bioinformatics* **25**, 1096-1098, doi:10.1093/bioinformatics/btp105 (2009).
- 7 Chen, L. *et al.* VFDB: a reference database for bacterial virulence factors. *Nucleic Acids Res* **33**, D325-328, doi:10.1093/nar/gki008 (2005).
- 8 Larsen, M. V. *et al.* Multilocus sequence typing of total-genome-sequenced bacteria. *J Clin Microbiol* **50**, 1355-1361, doi:10.1128/JCM.06094-11 (2012).
- 9 Jolley, K. A., Bray, J. E. & Maiden, M. C. J. Open-access bacterial population genomics: BIGSdb software, the PubMLST.org website and their applications. *Wellcome Open Res* **3**, 124, doi:10.12688/wellcomeopenres.14826.1 (2018).
- 10 Hawkey, J. *et al.* Global population structure and genotyping framework for genomic surveillance of the major dysentery pathogen, Shigella sonnei. *Nat Commun* **12**, 2684, doi:10.1038/s41467-021-22700-4 (2021).
- 11 National Health Commission of the People's Republic of China. *Announcement on the Issuance of 127 National Food Safety Standards, including the National Food Safety Standard for Fresh (Frozen) Livestock and Poultry Products (GB 2707-2016) (2016 No. 17)*, <<http://www.nhc.gov.cn/sps/s7891/201701/77b8a50e61c94522a589b925dc3b994f.shtml>> (2017).
- 12 National Health Commission of the People's Republic of China. *Standards for drinking water quality*, <<https://openstd.samr.gov.cn/bzgk/gb/newGbInfo?hcno=73D81F4F3615DDB2C5B1DD6BFC9DEC86>> (2006).
- 13 Moussatova, A., Kandt, C., O'Mara, M. L. & Tieleman, D. P. ATP-binding cassette transporters in Escherichia coli. *Biochim Biophys Acta* **1778**, 1757-1771,

- doi:10.1016/j.bbamem.2008.06.009 (2008).
- 14 Somers, W. S., Stahl, M. L. & Sullivan, F. X. GDP-fucose synthetase from *Escherichia coli*: structure of a unique member of the short-chain dehydrogenase/reductase family that catalyzes two distinct reactions at the same active site. *Structure* **6**, 1601-1612, doi:10.1016/s0969-2126(98)00157-9 (1998).
- 15 Agaisse, H. Molecular and Cellular Mechanisms of *Shigella flexneri* Dissemination. *Front Cell Infect Microbiol* **6**, 29, doi:10.3389/fcimb.2016.00029 (2016).
- 16 Koren, S. *et al.* Canu: scalable and accurate long-read assembly via adaptive k-mer weighting and repeat separation. *Genome Res* **27**, 722-736, doi:10.1101/gr.215087.116 (2017).
- 17 Walker, B. J. *et al.* Pilon: an integrated tool for comprehensive microbial variant detection and genome assembly improvement. *PLoS One* **9**, e112963, doi:10.1371/journal.pone.0112963 (2014).
- 18 Altschul, S. F., Gish, W., Miller, W., Myers, E. W. & Lipman, D. J. Basic local alignment search tool. *J Mol Biol* **215**, 403-410, doi:10.1016/s0022-2836(05)80360-2 (1990).
- 19 Ma, Q. *et al.* Multidrug-resistant *Shigella sonnei* carrying the plasmid-mediated mcr-1 gene in China. *Int J Antimicrob Agents* **52**, 14-21, doi:10.1016/j.ijantimicag.2018.02.019 (2018).
- 20 Liu, Y.-Y. *et al.* Emergence of plasmid-mediated colistin resistance mechanism MCR-1 in animals and human beings in China: a microbiological and molecular biological study. *The Lancet Infectious Diseases* **16**, 161-168, doi:10.1016/s1473-3099(15)00424-7 (2016).
- 21 Pham Thanh, D. *et al.* Inducible colistin resistance via a disrupted plasmid-borne mcr-1 gene in a 2008 Vietnamese *Shigella sonnei* isolate. *J Antimicrob Chemother* **71**, 2314-2317, doi:10.1093/jac/dkw173 (2016).
- 22 Ma, Q. *et al.* A Waterborne Outbreak of *Shigella sonnei* with Resistance to Azithromycin and Third-Generation Cephalosporins in China in 2015. *Antimicrob Agents Chemother* **61**, doi:10.1128/AAC.00308-17 (2017).
- 23 Mook, P. *et al.* ESBL-Producing and Macrolide-Resistant *Shigella sonnei* Infections among Men Who Have Sex with Men, England, 2015. *Emerg Infect Dis* **22**, 1948-1952, doi:10.3201/eid2211.160653 (2016).
- 24 WHO. Guidelines for the control of shigellosis, including epidemics due to *Shigella dysenteriae* type 1. Report No. 9241592330, 64 (2005).
- 25 Holt, K. E. *et al.* Tracking the establishment of local endemic populations of an emergent enteric pathogen. *Proc Natl Acad Sci U S A* **110**, 17522-17527, doi:10.1073/pnas.1308632110 (2013).
- 26 Baker, K. S. *et al.* Horizontal antimicrobial resistance transfer drives epidemics of multiple *Shigella* species. *Nat Commun* **9**, 1462, doi:10.1038/s41467-018-03949-8 (2018).
- 27 Baker, K. S. *et al.* Whole genome sequencing of *Shigella sonnei* through PulseNet Latin America and Caribbean: advancing global surveillance of foodborne illnesses. *Clin Microbiol Infect* **23**, 845-853, doi:10.1016/j.cmi.2017.03.021 (2017).
- 28 Chung The, H. *et al.* South Asia as a Reservoir for the Global Spread of Ciprofloxacin-Resistant *Shigella sonnei*: A Cross-Sectional Study. *PLoS Med* **13**, e1002055, doi:10.1371/journal.pmed.1002055 (2016).

- 29 Erdman, S. M., Buckner, E. E. & Hindler, J. F. Options for treating resistant *Shigella* species infections in children. *J Pediatr Pharmacol Ther* **13**, 29-43, doi:10.5863/1551-6776-13.1.29 (2008).
- 30 Muthuirulandi Sethuvel, D. P. *et al.* Phylogenetic and Evolutionary Analysis Reveals the Recent Dominance of Ciprofloxacin-Resistant *Shigella sonnei* and Local Persistence of *S. flexneri* Clones in India. *mSphere* **5**, doi:10.1128/mSphere.00569-20 (2020).
- 31 Pakbin, B., Didban, A., Brück, W. M. & Alizadeh, M. Phylogenetic analysis and antibiotic resistance of *Shigella sonnei* isolates. *FEMS Microbiol Lett* **369**, doi:10.1093/femsle/fnac042 (2022).